# Responsible AI (RAI) Games and Ensembles

**Yash Gupta**
Carnegie Mellon University
`yashgup2@cs.cmu.edu`

**Runtian Zhai**
Carnegie Mellon University
`rzhai@cs.cmu.edu`

**Arun Suggala**
Google Research
`arunss@google.com`

**Pradeep Ravikumar**
Carnegie Mellon University
`pradeepr@cs.cmu.edu`

## Abstract

Several recent works have studied the societal effects of AI; these include issues such as fairness, robustness, and safety. In many of these objectives, a learner seeks to minimize its worst-case loss over a set of predefined distributions (known as uncertainty sets), with usual examples being perturbed versions of the empirical distribution. In other words, aforementioned problems can be written as min-max problems over these uncertainty sets. In this work, we provide a general framework for studying these problems, which we refer to as *Responsible AI (RAI) games*. We provide two classes of algorithms for solving these games: (a) game-play based algorithms, and (b) greedy stagewise estimation algorithms. The former class is motivated by online learning and game theory, whereas the latter class is motivated by the classical statistical literature on boosting, and regression. We empirically demonstrate the applicability and competitive performance of our techniques for solving several RAI problems, particularly around subpopulation shift.

## 1 Introduction

In recent years, AI is increasingly being used in high-stakes decision-making contexts such as hiring, criminal justice, and healthcare. Given the impact these decisions can have on people's lives, it is important to ensure these AI systems have beneficial social effects. An emerging line of work has attempted to formalize such desiderata ranging over ethics, fairness, train-time robustness, test-time or adversarial robustness, and safety, among others. Each of these forms rich sub-fields with disparate desiderata, which are sometimes collated under the umbrella of "responsible AI". Many organizations are increasingly advocating the use of responsible AI models [Microsoft, 2021, Google, 2020].

But how do we do so when the majority of recent work around these problems is fragmented and usually focuses on optimizing one of these aspects at a time (DRO [Namkoong and Duchi, 2017, Duchi and Namkoong, 2018], GDRO [Sagawa et al., 2019], CVaR [Zhai et al., 2021a], Distribution Shift [Hashimoto et al., 2018, Zhai et al., 2021b])? Indeed optimizing for just one of these aspects has even been shown to exhibit adverse effects on the other aspects [Roh et al., 2020]. To address this, we study a general framework that is broadly applicable across many of the settings above, and which we refer to as *Responsible AI (RAI)* games. Our starting point is the recent understanding of a unifying theme in many of these disparate problems, that a learner seeks to minimize its worst-case loss over a set of predefined distributions. For example, in fairness, we seek to perform well on all sub-groups in the data. In robustness, we aim to design models that are robust to perturbations of the training data or the test distribution. This allows us to set up a zero-sum game between a learner that aims to learn a responsible model and an adversary that aims to prevent the learner from doing so. In

---

The relevant code for this work can be found at https://github.com/yashgupta-7/rai-games

the general RAI game setting, this is a computationally intractable game that need not even have a Nash equilibrium. To address this computational issue, we study a relaxation of the single predictor RAI game, which we term the *ensemble* RAI game, which can also be motivated as a linearization of the original RAI game.

We note that our framework encompasses not only the responsible AI settings but also the setting of classical boosting. Drawing upon the insights from boosting, we provide boosting-based algorithms for solving responsible AI games. We provide convergence guarantees of our algorithms by relying on the connections between boosting and online convex optimization, two-player gameplay [Arora et al., 2012, McMahan, 2011, Bubeck, 2011]. We also conduct empirical analyses to demonstrate the convergence and utility of our proposed algorithms. Interestingly, the algorithms allow for *plug-and-play* convenience, with changes in the RAI settings requiring only simple changes to the algorithms. More importantly, we could consider *intersections* of different responsible AI considerations, which in turn can simply be incorporated into our algorithms. Finally, we also study the population risks of our algorithms in certain important settings. We show a surprising result that for the case of binary classification with the $0/1$ loss, the optimal predictor for a large class of RAI games is the same as the Bayes optimal predictor, thus generalizing an emerging line of results demonstrating this for certain specific games [Hu et al., 2018]. Under such settings, solving the RAI game could nonetheless be helpful in finite sample settings (as also demonstrated in our experiments) since the RAI game serves to encode desiderata satisfied by the Bayes optimal classifier.

## 2 Problem Setup and Background

We consider the standard supervised prediction setting, with input random variable $X \in \mathcal{X} \subseteq \mathbb{R}^d$, output random variable $Y \in \mathcal{Y}$, and samples $S = \{(x_i, y_i)\}_{i=1}^n$ drawn from a distribution $P_{\text{data}}$ over $\mathcal{X} \times \mathcal{Y}$. Let $\widehat{P}_{\text{data}}$ denote the empirical distribution over the samples. We also have a set $H$ of hypothesis functions $h : \mathcal{X} \mapsto \mathcal{Y}$ from which we wish to learn the best predictor. We evaluate the goodness of a predictor via a loss function $\ell : \mathcal{Y} \times \mathcal{Y} \mapsto \mathbb{R}$, which yields the empirical risk: $\widehat{R}(h) = \mathbb{E}_{\widehat{P}_{\text{data}}} \ell(h(x), y)$ where $\mathbb{E}_{\widehat{P}_{\text{data}}}(f(x, y)) = \frac{1}{n} \sum_{i=1}^n f(x_i, y_i)$. Apart from having low expected risk, most settings require $h$ to have certain properties, for example, robustness to distribution shift, fairness w.r.t subpopulations, superior tail performance, resistance to adversarial attacks, robustness in the presence of outliers, etc. We cast all these subproblems into an umbrella term "Responsible AI". Each of these properties has been studied extensively in recent works, albeit individually. In this work, we attempt to provide a general framework to study these problems.

### 2.1 Related Work

We draw our unified framework from seminal works over the past decade by responsible AI researchers on devising non-expected risk objectives, *particularly min-max problems*, to ensure ML models are responsible. These have resulted in a multitude of different objectives (even for a single responsible AI desideratum such as fairness), and also multiple different sub-communities (so that fairness and multiple disparate robustness communities are relatively fractured), many (if not all) of which we combine within a single umbrella. There is emerging work on relating worst-case performance to invariance [Bühlmann, 2018]; in other words, we might be able to get approximate group invariance via minimizing an appropriately constructed worst-group risk and vice-versa.

**RAI aspects as constraints.** Many prior works have enforced robustness as a constrained optimization [Shafieezadeh-Abadeh et al., 2015, Gao and Kleywegt, 2022, Namkoong and Duchi, 2016, Ben-Tal et al., 2011]. There have also been few prior works enforcing fairness constraints [Mandal et al., 2020]. To the best of our knowledge, there exists minimal prior work focusing on multiple desiderata at once in this regard.

**Multi Objective Optimization.** Several works have considered a multi-objective view of ensuring fairness in classifiers [Martinez et al., 2020, Oneto et al., 2018]. If used for multiple RAI objectives, there is usually overhead in choosing a model that achieves a good trade-off between various losses. Also, it is difficult to guarantee that the solution is robust to any of the involved aspects. Our framework guarantees a certain level of performance on each of the RAI aspects under consideration.

**Distribution shift.** [Koh et al., 2021] classifies distribution shift problems into two categories: Domain generalization, and subpopulation shift. In this work, we focus on the subpopulation shift problem, where the target distribution is absolutely continuous to the source distribution. It has two

main applications: fairness [Hashimoto et al., 2018, Hu et al., 2018, Sagawa et al., 2019, Zhai et al., 2021b] and long-tail learning (*i.e.* learning on class-imbalanced datasets) [Cao et al., 2019, Menon et al., 2021, Kini et al., 2021].

**Distributionally Robust Optimization (DRO).** In DRO one aims to study classifiers that are robust to deviations of the data distribution. DRO has been studied under various uncertainty sets including $f$-divergence based uncertainty sets [Namkoong and Duchi, 2017, Duchi and Namkoong, 2018, Sagawa et al., 2019], Wasserstein uncertainty sets [Sinha et al., 2017, Gao et al., 2022], Maximum Mean Discrepancy uncertainty sets [Staib and Jegelka, 2019], more general uncertainty sets in the RKHS space [Zhu et al., 2020]. [Li et al., 2021a] evaluate model performance under worst-case subpopulations. Owing to its importance, several recent works have provided efficient algorithms for solving the DRO objective [Namkoong and Duchi, 2016, Qi et al., 2020, Kumar et al., 2023, Jin et al., 2021]. However, a lot of these techniques are specific to particular perturbation sets and are not directly applicable to the more general framework we consider in our work. Furthermore, in our work, we aim to learn an ensemble of models instead of a single model.

**Boosting.** Classical boosting aims to improve the performance of a weak learner by combining multiple weak classifiers to produce a strong classifier [Breiman, 1999, Friedman et al., 2000, Friedman, 2001, Freund and Schapire, 1995, Freund et al., 1996, Mason et al., 2000]. Over the years, a number of practical algorithms have been introduced such as AdaBoost [Schapire, 1999], LPBoost [Demiriz et al., 2002], gradient boosting [Mason et al., 1999], XGBoost [Chen and Guestrin, 2016], boosting for adversarial robustness [Zhang et al., 2022], [Meunier et al., 2021], [Balcan et al., 2023], and holistic robustness [Bennouna and Parys, 2022]. The algorithms we develop for RAI games are inspired by these algorithms.

**Fairness.** There are a number of fairness notions for algorithmic fairness, ranging from individual fairness [Dwork et al., 2012, Zemel et al., 2013], group fairness [Hardt et al., 2016a, Zafar et al., 2017], counterfactual fairness [Kusner et al., 2017], Rawlsian max-min fairness [Rawls, 2020, Hashimoto et al., 2018] and others [Barocas et al., 2017, Chouldechova and Roth, 2018, Mehrabi et al., 2021]. Our framework includes the popular notion of minimax group fairness. It doesn't capture other notions of group fairness such as Demographic Parity, Equality of Odds, Equality of Opportunity.

**Population RAI Risks.** Several recent works have studied properties of the population risks arising in various responsible AI scenarios. Hu et al. [2018] showed that the minimizer of population DRO risk (under general $f$-divergences) is the classical Bayes optimal classifier. Li et al. [2021b], Duchi and Namkoong [2018], Sinha et al. [2017] provided generalization guarantees for DRO risk under various divergence families ranging from $f$-divergences to Wasserstein perturbations.

## 3 RAI Games

In many cases, we do not wish to compute an unweighted average over training samples; due to reasons of noise, tail risk, robustness, and fairness, among many other "responsible AI" considerations.

**Definition 1 (RAI Risks)** *Given a set of samples $\{(x_i, y_i)\}_{i=1}^n$, we define the class of empirical RAI risks (for Responsible AI risks) as: $\widehat{R}_{W_n}(h) = \sup_{w \in W_n} \mathbb{E}_w(h(x), y)$, where $W_n \subseteq \Delta_n$, is some set of sample weights (a.k.a uncertainty set), and $\mathbb{E}_w(f(x, y)) = \sum_{i=1}^n w_i f(x_i, y_i)$.*

Various choices of $W_n$ give rise to various RAI risks. Table 1 presents examples of RAI risks that are popular in ML. Interestingly, classical problems such as boosting are special cases of RAI risks. In this work, we rely on this connection to design boosting-inspired algorithms for minimizing RAI risks. More choices for $W_n$ can be obtained by combining the one's specified in Table 1 using union, intersection, convex-combination operations. For example, if one wants models that are fair to certain pre-specified groups, and at the same-time achieve good tail-risk, then one could choose $W_n$ to be the intersection of Group-DRO and $\alpha$-CVaR uncertainty sets.

Given the empirical RAI risk $\widehat{R}_{W_n}(h)$ of a hypothesis, and set of hypotheses $H$, we naturally wish to obtain the hypothesis that minimizes the empirical RAI risk: $\min_{h \in H} \widehat{R}_{W_n}(h)$. This can be seen as solving a zero-sum game.

**Definition 2 (RAI Games)** *Given a set of hypothesis $H$, and a RAI sample weight set $W_n$, the class of RAI games is given as: $\min_{h \in H} \max_{w \in W_n} \mathbb{E}_w(h(x), y)$.*

We thus study RAI Games for the special cases above and for an arbitrary constraint set $W_n$.

| Name | $W_n$ | Description |
|---|---|---|
| Empirical Risk Minimization | $\{\widehat{P}_{\text{data}}\}$ | object of focus in most of ML/AI |
| Worst Case Margin | $\Delta_n$, 
 entire probability simplex | used for designing 
 margin-boosting algorithms 
 [Warmuth et al., 2006, Bartlett et al., 1998] |
| Soft Margin | $\{w : KL(w\|\|\widehat{P}_{\text{data}}) \leq \rho_n\}$ | used in the design of 
 AdaBoost [Freund and Schapire, 1995] |
| $\alpha$-Conditional Value 
 at Risk (CVaR) | $\{w : w \in \Delta_n, w \preceq \frac{1}{\alpha n}\}$ | used in fairness 
 [Zhai et al., 2021a, Sagawa et al., 2019] |
| Distributionally Robust 
 Optimization (DRO) | $\{w : D(w\|\|\widehat{P}_{\text{data}}) \leq \rho_n\}$ | various choices for $D$ 
 have been studied 
 $f$-divergence [Duchi and Namkoong, 2018] |
| Group DRO | $\{\widehat{P}_{\text{data}}(G_1), \widehat{P}_{\text{data}}(G_2), \dots \widehat{P}_{\text{data}}(G_K)\}$ 
 $\widehat{P}_{\text{data}}(G_i)$ is dist. of $i^{th}$ group | used in group fairness, agnostic 
 federated learning [Mohri et al., 2019] |

Table 1: Various ML/AI problems that fall under the umbrella of RAI risks.

# 4 Ensemble RAI Games

In this section, we begin our discussion about ensembles. In general, a statistical caveat with Definition 2 is that good worst-case performance over the sample weight set $W_n$ is generally harder, and for a simpler set of hypotheses $H$, there may not exist $h \in H$ that can achieve such good worst-case performance. Thus it is natural to consider deterministic ensemble models over $H$, which effectively gives us more powerful hypothesis classes. Let us first define RAI risk for such classifiers.

**Definition 3 (Deterministic Ensemble)** *Consider the problem of classification, where $\mathcal{Y}$ is a discrete set. Given a hypothesis class $H$, a deterministic ensemble is specified by some distribution $Q \in \Delta_H$, and is given by: $h_{det;Q}(x) = \arg\max_{y \in \mathcal{Y}} \mathbb{E}_{h \sim Q}\mathbb{I}[h(x) = y]$. Correspondingly, we can write the deterministic ensemble RAI risk as $\widehat{R}_{W_n}(h_{det;Q}(x)) = \max_{w \in W_n} \mathbb{E}_w \ell(h_{det;Q}(x), y)$.*

We discuss alternative definitions of deterministic ensembles in the Appendix. This admits a class of deterministic RAI games:

**Definition 4 (Deterministic Ensemble RAI Games)** *Given a set of hypothesis $H$, a RAI sample weight set $W_n$, the class of RAI games for deterministic ensembles over $H$ is given as:*

$$\min_{Q \in \Delta_H} \max_{w \in W_n} \mathbb{E}_w \ell(h_{det;Q}(x), y).$$

However, the aforementioned game is computationally less amenable because of the non-smooth nature of de-randomized predictions. Moreover, there are some broader challenges with RAI games given by Definitions 2 and 4. Firstly, they need not have a Nash Equilibrium (NE), and in general, their min-max and max-min game values need not coincide. This poses challenges in solving the games efficiently. Next, in some cases, directly optimizing over the worst-case performance might not even be useful. For instance, [Hu et al., 2016, Zhai et al., 2021a] show the pessimistic result that for classification tasks where when models are evaluated by the zero-one loss, ERM achieves the lowest possible DRO loss defined by some $f$-divergence or the $\alpha$-CVaR loss, given that the model is deterministic. To this end, we consider the following randomized ensemble:

**Definition 5 (Randomized Ensemble)** *Given a hypothesis class $H$, a randomized ensemble is specified by some distribution $Q \in \Delta_H$, and is given by: $\mathbb{P}[h_{rand;Q}(x) = y] = \mathbb{E}_{h \sim Q}\mathbb{I}[h(x) = y]$. Similarly, we can define its corresponding randomized ensemble RAI risk: $\widehat{R}_{rand;W_n}(Q) = \max_{w \in W_n} \mathbb{E}_{h \sim Q}\mathbb{E}_w \ell(h(x), y)$.*

We can then also define the class of ensemble RAI games:

**Definition 6 (Randomized Ensemble RAI Games)** *Given a set of hypothesis $H$, a RAI sample weight set $W_n$, the class of mixed RAI games is given as:*

$$\min_{Q \in \Delta_H} \max_{w \in W_n} \mathbb{E}_{h \sim Q}\mathbb{E}_w \ell(h(x), y). \tag{1}$$

This is a much better class of zero-sum games: it is linear in both the hypothesis distribution $P$, as well as the sample weights $w$, and if the sample weight set $W_n$ is convex, is a convex-concave game. As shown below, under some mild conditions, this game has a Nash equilibrium which can be well approximated via efficient algorithms.

**Proposition 1** *Let $H$ be parameterized by $\theta \in \Theta \subseteq \mathbb{R}^p$, for convex, compact set $\Theta$ and let $W_n$ be a convex, compact set. Then $\min_{Q \in \Delta_H} \max_{w \in W_n} \mathbb{E}_{h \sim Q} \mathbb{E}_w \ell(h(x), y) = \max_{w \in W_n} \min_{Q \in \Delta_H} \mathbb{E}_{h \sim Q} \mathbb{E}_w \ell(h(x), y)$*

The proposition follows as a direct consequence of well known minimax theorems (Appendix D.3).

### 4.1 Going from Deterministic to Randomized Ensembles

To begin, we point out that what we want is a deterministic ensemble rather than a randomized ensemble. In fact, it can be seen that the deterministic ensemble in Definition 3 is a specific de-randomization of the randomized ensemble. It is such deterministic ensembles that we usually simply refer to as ensemble predictors. But the RAI risk for the ensemble predictor is NOT equal to the ensemble RAI risk minimized by our desired game in Equation 1 above for randomized ensembles. Thus, the ensemble RAI game might not in general capture the ideal deterministic ensemble. In this section, we study why and when might solving for a random ensemble is meaningful.

**Binary Classification.** Interestingly, for the very specific case of binary classification, we can provide simple relationships between the risks of the randomized and deterministic ensemble.

**Proposition 2** *Consider the setting with $\mathcal{Y} = \{-1, 1\}$, the zero-one loss $\ell$, and $W_n = \Delta_n$. Then,*
$$\widehat{R}_{W_n}(h_{det;Q}) = \mathbb{I}[\widehat{R}_{W_n}(h_{rand;Q}) \geq 1/2].$$

See Appendix E.2 for a simple proof. In this case, we can also relate the existence of a perfect deterministic ensemble ("boostability") to a weak learning condition on the set of hypotheses. Specifically, suppose $H$ is boostable iff there exists $Q \in \Delta_H$ s.t. $\widehat{R}_{W_n}(h_{\text{det};Q}) = 0$. From the above proposition this is equivalent to requiring that $\widehat{R}_{W_n}(h_{\text{rand};Q}) < 1/2$. We thus obtain:
$$\inf_{Q \in \Delta_H} \sup_{w \in W_n} \mathbb{E}_{w,Q} \ell(h(x), y) < 1/2 \iff \sup_{w \in W_n} \inf_{h \in H} \mathbb{E}_w \ell(h(x), y) < 1/2$$
where the equivalence follows from the min-max theorem and the linearity of the objective in $P$. The last statement says that for any sample weights $w \in W_n$, there exists a hypothesis $h \in H$ that has $w$-weighted loss at most $1/2$. We can state this as a "weak-learning" condition on individual hypotheses in $H$. The above thus shows that for the specific case of $\mathcal{Y} = \{-1, 1\}$, the zero-one loss $\ell(y, y') = \mathbb{I}[y \neq y']$, and $W_n = \Delta_n$, we can relate boostability of $H$ to a weak learning condition on hypothesis within $H$.

**General Classification** But in general, we do not have simple connections between $\widehat{R}_{W_n}(h_{\text{det};Q})$ and $\widehat{R}_{W_n}(h_{\text{rand};Q})$. All we can guarantee is the following upper bound:

**Proposition 3** *Let $\gamma_Q = 1/\min_{i \in [n]} \max_{y \in \mathcal{Y}} \mathbb{P}_Q[h(x_i) = y]$. Then,*
$$\widehat{R}_{W_n}(h_{det;Q}) \leq \gamma_Q \widehat{R}_{W_n}(h_{rand;Q}).$$

See Appendix E.2 for a simple proof.

**Corollary 4** *For binary classification, we have $\gamma_P \leq 2$ and thus, we recover the well known bound $\widehat{R}_{W_n}(h_{det;Q}) \leq 2\widehat{R}_{W_n}(h_{rand;Q})$*

**Remark 5** *These bounds might be loose in practice. Specifically, for the binary case, if $\widehat{R}_{W_n}(h_{rand;Q}) \leq \frac{1}{2}$ then we have $\widehat{R}_{W_n}(h_{det;Q}) = 0$. To this end, prior work [Lacasse et al., 2006, Germain et al., 2015, Masegosa et al., 2020] have developed tighter bounds using second-order inequalities. We leave the analyses of these second-order RAI games to future work.*

As such, we can cast minimizing randomized RAI risk as minimizing an upper bound on the deterministic ensemble RAI risk. Thus, the corresponding randomized RAI game can be cast as a relaxation of the deterministic RAI game. In the sequel, we thus focus on this randomized ensemble RAI game, which we will then use to obtain a deterministic ensemble. Following the bounds above, the corresponding deterministic ensemble risk will be bounded by the randomized ensemble RAI risk

## 5 Algorithms

In this section, we present two algorithms for solving the RAI game in Equation (1). Our first algorithm is motivated from online learning algorithms and the second algorithm is motivated from

greedy stepwise algorithms that have been popular for solving many statistical problems such as regression. For simplicity of presentation, we assume $H$ is a finite set. However, our results in the section extend to uncountable sets.

## 5.1 Methods

**Game-play.** In game play based algorithms, both the min and the max players are engaged in a repeated game against each other. Both players rely on no-regret algorithms to decide their next action. It is well known that such a procedure converges to a mixed NE of the game Cesa-Bianchi and Lugosi [2006]. In this work, we follow a similar strategy to solve the game in Equation (1) (see Algorithm 1 for the pseudocode). In the $t^{th}$ round of our algorithm, the following distribution $w^t \in W$ is computed over the training data points

$$w^t \leftarrow \underset{w \in W_n}{\operatorname{argmax}} \sum_{s=1}^{t-1} \mathbb{E}_w \ell(h^s(x), y) + \eta^{t-1} \operatorname{Reg}(w) \tag{2}$$

This update is called the Follow-The-Regularized-Leader (FTRL) update. Here, $\operatorname{Reg}(\cdot)$ is a strongly concave regularizer and $\eta^{t-1}$ is the regularization strength. One popular choice for $\operatorname{Reg}(\cdot)$ is the negative entropy which is given by $-\sum_i w_i \log w_i$. This regularizer is also used by AdaBoost, which is a popular boosting algorithm. In Appendix F.2, we provide analytical expressions for $w^t$ for various choices of $W_n, \operatorname{Reg}(\cdot)$. We note that the regularizer in the FTRL update ensures the stability of the updates; *i.e.,* it ensures consecutive iterates do not vary too much. This stability is naturally guaranteed when $W_n$ is a strongly convex set (an example of a strongly convex set is the level set of a strongly convex function. See Appendix for a formal definition and more details). Consequently, the regularization strength $\eta^{t-1}$ could be set to $0$ in this case, and the algorithm still converges to a NE [Huang et al., 2017].

Once we have $w^t$, a new classifier $h^t$ is computed to minimize the weighted loss relative to $w^t$, and added to the ensemble. This update is called the Best Response (BR) update. Learning $h^t$ in this way helps us fix past classifiers' mistakes, eventually leading to an ensemble with good performance.

---

**Algorithm 1** Game play algorithm for solving Equation (1)

> **Input:** Training data $\{(x_i, y_i)\}_{i=1}^n$, loss function $\ell$, constraint set $W_n$, hypothesis set $H$, strongly concave regularizer $R$ over $W_n$, learning rates $\{\eta^t\}_{t=1}^T$
> 1: **for** $t \leftarrow 1$ to $T$ **do**
> 2:     **FTRL**: $w^t \leftarrow \operatorname{argmax}_{w \in W_n} \sum_{s=1}^{t-1} \mathbb{E}_w \ell(h^s(x), y) + \eta^{t-1} \operatorname{Reg}(w)$
> 3:     **BR**: $h^t \leftarrow \operatorname{argmin}_{h \in H} \mathbb{E}_{w^t} \ell(h(x), y)$
> 4: **end for**
> 5: **return** $P^T = \frac{1}{T} \sum_{t=1}^T w^t$, $Q^T = \operatorname{Unif}\{h^1, \ldots h^T\}$

---

**Greedy.** We now take an optimization theoretic viewpoint to design algorithms for Equation (1). Let $L(Q)$ denote the inner maximization problem of (1): $L(Q) \coloneqq \max_{w \in W_n} \mathbb{E}_{h \sim Q} \mathbb{E}_w \ell(h(x), y)$. When $L(Q)$ is smooth (this is the case when $W_n$ is a strongly convex set), one could use Frank-Wolfe (FW) to minimize it. The updates of this algorithm are given by

$$Q^t \leftarrow (1 - \alpha^t) Q^{t-1} + \alpha^t G, \quad \text{where } G = \underset{Q}{\operatorname{argmin}} \left\langle Q, \nabla_Q L(Q^{t-1}) \right\rangle.$$

Here, $\nabla_Q L(Q^{t-1}) = \operatorname{argmax}_{w \in W_n} \mathbb{E}_{h \sim Q^{t-1}} \mathbb{E}_w \ell(h(x), y)$. This algorithm is known to converge to a minimizer of $L(Q)$ at $O(1/t)$ rate [Jaggi, 2013]. When $L(Q)$ is non-smooth, we first need to smooth the objective before performing FW. In this work we perform Moreau smoothing [Parikh et al., 2014], which is given by

$$L_\eta(Q) = \max_{w \in W_n} \mathbb{E}_{h \sim Q} \mathbb{E}_w \ell(h(x), y) + \eta \operatorname{Reg}(w). \tag{3}$$

Here $\operatorname{Reg}(\cdot)$ is a strongly concave regularizer. If $\operatorname{Reg}(\cdot)$ is 1-strongly concave, it is well known that $L_\eta(Q)$ is $O(1/\eta)$ smooth. Once we have the smoothed objective, we perform FW to find its optimizer (see Algorithm 2 for pseudocode).

**Relaxing the simplex constraint.** We now derive a slightly different algorithm by relaxing the simplex constraint on $Q$. Using Lagrangian duality we can rewrite $\min_{Q \in \Delta_H} L_\eta(Q)$ as the following problem for some $\lambda \in \mathbb{R}$

$$\min_{Q \succeq 0} L_\eta(Q) + \lambda \sum_{h \in H} Q(h).$$

One interesting observation is that when $W_n$ is the entire simplex and when $\lambda = -1/2$, we recover the AdaBoost algorithm. Given the practical success of AdaBoost, we extend it to general $W_n$. In particular, we set $\lambda = -1/2$ and solve the resulting objective using greedy coordinate-descent. The updates of this algorithm are given in Algorithm 2.

**Remark 6** *Algorithm 2 takes the step sizes $\{\alpha^t\}_{t=1}^T$ as input. In practice, one could use line search to figure out the optimal step-sizes, for better performance.*

---

**Algorithm 2** Greedy algorithms for solving Equation (1)

---

**Input:** Training data $\{(x_i, y_i)\}_{i=1}^n$, loss function $\ell$, constraint set $W_n$, hypothesis set $H$, strongly concave regularizer $R$ over $W_n$, regularization strength $\eta$, step sizes $\{\alpha^t\}_{t=1}^T$
1: **for** $t \leftarrow 1$ to $T$ **do**
2:     $G^t = \text{argmin}_Q \langle Q, \nabla_Q L_\eta(Q^{t-1}) \rangle$
3:     **FW:** $Q^t \leftarrow (1 - \alpha^t)Q^{t-1} + \alpha^t G^t$ / **Gen-AdaBoost:** $Q^t \leftarrow Q^{t-1} + \alpha^t G^t$
4: **end for**
5: **return** $Q^T$

---

We provide convergence rates for the algorithms below:

**Proposition 7** *(Convergence Rates) Let $l(h(x), y) \in [0,1] \ \forall h \in H, (x,y) \in D$ and $Reg : \Delta_n \to \mathbb{R}$ be a 1-strongly concave function w.r.t norm $\|.\|_1$. Let $Q^T$ be the output returned from running Algorithm 1 or 2 for $T$ iterations. Let $D_R$ be a constant S.T. $D_R^2 = \max_{x,y \in W_n} |Reg(x) - Reg(y)|$.*

    1. *(**Gameplay**) If $\eta^t = \eta$, then $Q^T$ satisfies $L(Q^T) \leq \min_Q L(Q) + \frac{\eta D_R^2}{T} + \mathcal{O}(\frac{1}{\eta})$.*

    2. *(**Greedy**) If line-search is performed for $\alpha^t$, then $Q^T$ (FW or the Gen-AdaBoost update) satisfies $L(Q^T) \leq \min_Q L(Q) + \eta D_R^2 + \mathcal{O}(\frac{1}{\eta T})$.*

We refer the reader to Appendix F.1 for a simple proof using existing theory on online convex optimization [McMahan, 2011, Jaggi, 2013]. Another useful insight is that Algorithms 1 and 2 are related to each other under special settings as shown by Appendix H.1.

**Corollary 8** *Consider $Reg(w) = -\sum_{i=1}^n w_i \log w_i$ and $l$ as the zero-one loss. Then, Algorithm 1 and Algorithm 2 (line-search) achieve $\epsilon-$approximate NE with $\epsilon$ as $\mathcal{O}\left(\sqrt{\frac{\log(n)}{T}}\right)$.*

**Weak Learning Conditions** It might not be practical for $H$-player to play BR (Step 3: Algorithm 1) or correspondingly, to find the best possible classifier at every round (Step 2: Algorithm 2). Under weak learning conditions, we can indeed achieve (approximate) convergence when we only solve these problems approximately. See Appendix H.2 for more details.

## 6 Generalization Guarantees

In this section, we study the population RAI risk and present generalization bounds which quantify the rates at which empirical RAI risk converges to its population counterpart.

### 6.1 Population RAI Games

Recall, the empirical RAI risk optimizes over all sample re-weightings $w \in W_n$ that lie within the probability simplex $\Delta_n$. Thus it's population counterpart optimizes over distributions $P$ that are absolutely continuous with respect to the data distribution $P_{\text{data}}$:

$$R_W(h) = \sup_{P : P \ll P_{\text{data}}, \frac{dP}{dP_{\text{data}}} \in W} \mathbb{E}_P[\ell(h(x), y)].$$

Following [Shapiro et al., 2021], we can rewrite this as follows. Suppose we use $Z = (X, Y) \in \mathcal{Z} := \mathcal{X} \times \mathcal{Y}$, so that $P$, $P_{\text{data}}$ are distributions over $Z$. We then define $\ell_h : \mathcal{Z} \mapsto \mathbb{R}$ as $\ell_h(z) = \ell(h(x), y)$. We can then write the population RAI risk as (see Appendix G for a proof):

$$R_W(h) = \sup_{r: \mathcal{Z} \mapsto \mathbb{R}_+, \int r(z) dP_{\text{data}}(z) = 1, r \in W} \mathbb{E}_{P_{\text{data}}}[r(z) \ell_h(z)]. \tag{4}$$

For classification, we define the RAI-Bayes optimal classifier as: $Q_W^* = \arg\min_Q R_W(Q)$. Here, the minimum is w.r.t the set of all measurable classifiers (both deterministic and random). This is the "target" classifier we wish to learn given finite samples. Note that this might not be the same as the vanilla Bayes optimal classifier: $Q^* = \arg\min_Q \mathbb{E}[\widehat{R}(Q)]$, which only minimizes the expected loss, and hence may not satisfactorily address RAI considerations.

We now try to characterize the RAI-Bayes optimal classifier. However, doing this requires a bit more structure on $W$. So, in the sequel, we consider constraint sets of the following form:

$$W = \left\{ r : \mathcal{Z} \mapsto \mathbb{R}_+ \; : \; \int g_i(r(z)) dP_{\text{data}}(z) \le c_i, i \in [m] \right\}, \tag{5}$$

where we assume that $g_i : \mathbb{R}_+ \mapsto \mathbb{R}, i \in [m]$ are convex. Note that this choice of $W$ encompasses a broad range of RAI games including DRO with $f$-divergence, CVaR, soft-margin uncertainty sets. Perhaps surprisingly, the following proposition shows that the minimizer of population RAI risk is nothing but the vanilla Bayes optimal classifier.

**Proposition 9 (Bayes optimal classifier)** *Consider the problem of binary classification where $\mathcal{Y} = \{-1, +1\}$. Suppose $\ell(h(x), y) = \phi(yh(x))$ for some $\phi : \mathbb{R} \to [0, \infty)$ which is either the 0/1 loss, or a convex loss function that is differentiable at 0 with $\phi'(0) < 0$. Suppose the uncertainty set $W$ is as specified in Equation (5). Moreover, suppose $\{g_i\}_{i=1...m}$ are convex and differentiable functions. Then, the vanilla Bayes optimal classifier is also a RAI-Bayes optimal classifier.*

**Remark 10** *In the special case of $m = 1$ in Equation (5), we recover the result of [Hu et al., 2018]. However, our proof is much more elegant than the proof of [Hu et al., 2018], and relies on the dual representation of the population RAI risk.*

One perspective of the above result is that the vanilla Bayes optimal classifier is also "responsible" as specified by the RAI game. This is actually reasonable in many practical prediction problems where the label annotations are actually derived from humans, who presumably are also responsible. Why then might we be interested in the RAI risk? One advantage of the RAI risks is in finite sample settings where the equivalence no longer holds, and the RAI risk could be construed as encoding prior knowledge about properties of the Bayes optimal classifier. We also note that the above equivalence is specific for binary classification.

## 6.2 Generalization Guarantees

Our generalization bounds rely on the following dual characterization of the RAI population risk.

**Proposition 11** *Suppose the uncertainty set $W$ is as specified in Equation (5). Then for any hypothesis $h$, the population RAI risk can be equivalently written as*

$$R_W(h) = \inf_{\lambda \ge \mathbf{0}, \tau} \mathbb{E}_{P_{\text{data}}} G_\lambda^*(\ell_h(z) - \tau) + \sum_{i=1}^m \lambda_i c_i + \tau, \tag{6}$$

*where $G_\lambda^*$ is the Fenchel conjugate of $G_\lambda(t) = \sum_{i=1}^m \lambda_i g_i(t)$.*

We utilize the above expression for $R_W(h)$ to derive the following deviation bound for $\widehat{R}_{W_n}(h)$.

**Theorem 12** *Consider the setting of Proposition 11. Suppose $\{g_i\}_{i=1...m}$ are convex and differentiable functions. Suppose $\ell_h(z) \in [0, B]$ for all $h \in H$, $z \in \mathcal{Z}$. Suppose, for any distribution $P_{\text{data}}$, the minimizers $(\lambda^*, \tau^*)$ of Equation (6) lie in the following set: $\mathcal{E} = \{(\lambda, \tau) : \|\lambda^*\|_\infty \le \bar{\Lambda}, |\tau^*| \le T\}$. Moreover, let's suppose the optimal $\lambda^*$ for $P_{\text{data}}$ is bounded away from 0 and satisfies $\min_i \lambda_i^* \ge \underline{\Lambda}$.*

*Let $G, L$, be the range and Lipschitz constants of $G_\lambda^*$:*

$$G := \sup_{(\lambda, \tau) \in \mathcal{E}} G_\lambda^*(B - \tau) - G_\lambda^*(-\tau), \quad L := \sup_{x \in [0, B], (\lambda, \tau) \in \mathcal{E}, \lambda: \min_i \lambda_i \ge \underline{\Lambda}} \left\| \frac{\partial G_\lambda^*(x - \tau)}{\partial(\lambda, \tau)} \right\|_2.$$

*For any fixed $h \in H$, with probability at least $1 - 2e^{-t}$*

$$|R_W(h) - \widehat{R}_{W_n}(h)| \leq 10n^{-1/2}G(\sqrt{t + m\log(nL)}).$$

Given Theorem 12, one can take a union bound over the hypothesis class $H$ to derive the following uniform convergence bounds.

**Corollary 13** *Let $N(H, \epsilon, \|\cdot\|_{L^\infty(\mathcal{Z})})$ be the covering number of $H$ in the sup-norm which is defined as $\|h\|_{L^\infty(\mathcal{Z})} = \sup_{z \in \mathcal{Z}} |h(z)|$. Then with probability at least $1 - N(H, \epsilon_n, \|\cdot\|_{L^\infty(\mathcal{Z})})e^{-t}$, the following holds for any $h \in H$: $|R_W(h) - \widehat{R}_{W_n}(h)| \leq 30n^{-1/2}G(\sqrt{t + m\log(nL)})$. Here $\epsilon_n = n^{-1/2}G\sqrt{t + m\log(nL)}$.*

The above bound depends on parameters $(\lambda^*, \tau^*, G, L)$ which specific to the constraint set $W$. To instantiate it for any $W$ one needs to bound these parameters. We note that our generalization guarantees become sub-optimal as $\underline{\lambda} \to 0$. This is because the Lipschitz constant $L$ could potentially get larger as $\lambda$ approaches the boundary. Improving these bounds is an interesting future direction.

**Remark 14** *We note that aforementioned results results follow from relatively stringent assumptions. Exploring the impact of relaxing these assumptions is an interesting direction for future works.*

# 7  Experiments

In this section, we demonstrate the generality of proposed RAI methods by studying one of the most well-studied problems in RAI i.e. the case of subpopulation shift. Given a large number of possible $W$, we acknowledge that this is not a complete analysis, even with respect to the problems that live within the minimax framework. Instead, we aim to display convergence, `plug-and-play` generality, and superior performance over some seminal baselines of this task. We conduct experiments on both synthetic and real-world datasets. Please refer to Appendix for details on synthetic experiments. We consider a number of responsible AI settings, including subpopulation shift, in the domain-oblivious (DO) setting where we do not know the sub-populations [Hashimoto et al., 2018, Lahoti et al., 2020, Zhai et al., 2021a], the domain-aware (DA) setting where we do [Sagawa et al., 2019], and the partially domain-aware (PDA) setting where only some might be known.

**Datasets & Domain Definition.** We use the following datasets: COMPAS [Angwin et al., 2016], CIFAR-10 (original, and with a class imbalanced split [Jin et al., 2021, Qi et al., 2021]) and CIFAR-100. See the Appendix for more details on our datasets. For COMPAS, we consider *race (White vs Other)* and *biological gender (Male vs Female)* as our sensitive attributes. This forms four disjoint subgroups defined by these attributes. In the PDA setting, we partition only across the attribute *race* while training, but still run tests for all four subgroups. On CIFAR-10, class labels define our 10 subpopulations. Similarly as above, for the PDA setting, we make 5 super-groups of two classes each. On CIFAR-100, class labels define our 100 subpopulations. For the PDA setting, we make 20 super-groups, each consisting of five classes.

**Baselines.** We compare our method against the following baselines: (a) Deterministic classifiers trained on empirical risk (ERM) and DRO risks, particularly the quasi-online algorithm for Group DRO [Sagawa et al., 2019] (Online GDRO), and an ITLM-inspired SGD algorithm [Zhai et al., 2021b, Shen and Sanghavi, 2018] for $\chi^2$ DRO (SGD ($\chi^2$)) (b) Ensemble models AdaBoost [Schapire, 1999]. Note that the purpose of our experiments is to show that we can match baselines for a specific single desideratum (e.g. worst-case sub-population) while allowing for learning models that can solve multiple responsible AI desiderata at the same time, for which we have no existing baselines.

**Proposed Methods.** We focus on Algorithm 2 and refer to FW and Gen-AdaBoost updates as RAI-FW and RAI-GA, respectively. Moreover, our implementations include the following alterations: • We track the unregularized objective value from Equation 1 for the validation set, and whenever it increases we double the regularization factor $\eta$, which we find can improve generalization. • We also use this objective w.r.t the normalized $Q^t$ to perform a line search for the step size $\alpha$. For the FW update, our search space is a ball around $\frac{1}{t}$ at round $t$, while for GA, we search within $(0, 1)$.

**Base Learners & Training.** Training time scales linearly with the number of base learners. Inference, though, can be parallelized if need be. We usually find training on 3-5 learners is good enough on all scenarios explored in the paper. We defer further details of our base learners and hyperparameter choices to the Appendix.

**Constraint sets $W_n$.** For RAI algorithms, we use the following constraint sets: • Domain Oblivious (DO): We use the $\chi^2$-DRO constraint set to control for worst-case subpopulations. • Domain Aware (DA): We use the Group DRO constraint set as the domain definitions are known. • Partially Domain-Aware (PDA): We use a novel set $W_n$ which is the intersection over Group DRO constraints over the known domains and $\chi^2$ constraints to control for unknown group performance. For baselines, we use AdaBoost and SGD($\chi^2$) for the DO setting. Online GDRO serves as our baseline for both DA and PDA settings, where the algorithm uses whatever domain definitions are available.

Table 2: (Table 1 in the paper) Mean and worst-case expected loss for baselines, RAI-GA and RAI-FW. (Complex) indicates the use of larger models. Constraint sets $W_n$ are indicated in (.). Each experiment is carried out over three random seeds and confidence intervals are reported.

| Setting | Algorithm | COMPAS | | CIFAR-10 (Imbalanced) | | CIFAR10 | | CIFAR100 | |
|---|---|---|---|---|---|---|---|---|---|
| | | Average | Worst Group | Average | Worst Class | Average | Worst Class | Average | Worst Class |
| DO (Complex) | ERM | 31.3 ±0.2 | 31.7 ±0.1 | 12.1 ±0.3 | 30.4 ±0.2 | 8.3 ±0.2 | 21.3 ±0.5 | 25.2 ±0.2 | 64.0 ±0.7 |
| | RAI-GA ($\chi^2$) | 31.3 ±0.2 | **31.2** ±0.2 | 11.7 ±0.4 | **29.0** ±0.3 | 8.2 ±0.1 | 19.0 ±0.1 | 25.6 ±0.4 | **56.8** ±0.8 |
| | RAI-FW ($\chi^2$) | 31.2 ±0.1 | 31.4 ±0.3 | 11.9 ±0.1 | 29.1 ±0.2 | 8.0 ±0.3 | **15.4** ±0.4 | 25.4 ±0.2 | 58.0 ±1.1 |
| DO | ERM | 32.1 ±0.3 | 34.6 ±0.4 | 14.2 ±0.1 | 33.6 ±0.3 | 11.4 ±0.4 | 27.0 ±0.1 | 27.1 ±0.3 | 66.0 ±1.1 |
| | AdaBoost | 31.8 ±0.4 | 32.6 ±0.3 | 15.2 ±0.2 | 40.6 ±0.2 | 12.0 ±0.1 | 28.7 ±0.3 | 28.1 ±0.2 | 72.2 ±1.2 |
| | SGD ($\chi^2$) | 32.0 ±0.2 | 33.7 ±0.2 | 13.3 ±0.3 | 31.7 ±0.4 | 11.3 ±0.3 | 24.7 ±0.1 | 27.4 ±0.1 | 65.9 ±1.2 |
| | RAI-GA ($\chi^2$) | 31.5 ±0.2 | 33.2 ±0.3 | 14.0 ±0.1 | 32.2 ±0.2 | 10.8 ±0.4 | 25.0 ±0.2 | 27.4 ±0.4 | 65.0 ±0.8 |
| | RAI-FW ($\chi^2$) | 31.6 ±0.1 | **32.5** ±0.5 | 13.9 ±0.1 | 32.6 ±0.3 | 10.9 ±0.4 | **23.4** ±0.2 | 27.5 ±0.1 | **63.8** ±0.6 |
| DA | Online GDRO | 31.7 ±0.2 | 32.2 ±0.3 | 13.1 ±0.2 | 26.6 ±0.2 | 11.2 ±0.1 | 21.7 ±0.3 | 27.3 ±0.1 | 57.0 ±0.5 |
| | RAI-GA (Group) | 32.0 ±0.1 | 32.7 ±0.1 | 13.0 ±0.3 | 27.3 ±0.4 | 11.5 ±0.1 | 22.4 ±0.2 | 27.4 ±0.2 | 56.6 ±1.1 |
| | RAI-FW (Group) | 32.1 ±0.2 | 32.3 ±0.2 | 13.0 ±0.2 | **26.0** ±0.1 | 11.4 ±0.3 | **20.3** ±0.1 | 27.9 ±0.2 | **52.9** ±0.9 |
| PDA | Online GDRO | 31.5 ±0.1 | 32.7 ±0.2 | 13.4 ±0.1 | 32.2 ±0.2 | 11.3 ±0.2 | 25.2 ±0.1 | 27.7 ±0.2 | 64.0 ±0.8 |
| | RAI-GA (Group ∩ $\chi^2$) | 31.4 ±0.4 | 32.9 ±0.2 | 13.0 ±0.3 | 30.1 ±0.1 | 10.8 ±0.2 | **23.7** ±0.2 | 27.5 ±0.1 | 62.5 ±0.6 |
| | RAI-FW (Group ∩ $\chi^2$) | 31.8 ±0.2 | **32.3** ±0.1 | 13.5 ±0.3 | **29.4** ±0.3 | 11.2 ±0.4 | 24.0 ±0.2 | 27.9 ±0.3 | **58.9** ±0.7 |

**Results and Discussion.** We run our methods and baselines under the settings described above and report the results in Table 2. As such, we can make the following observations:

1. RAI-FW and RAI-GA methods significantly improve the worst-case performance with only a few base learners across all datasets in all three settings, while maintaining average case performance. Moreover, For seemingly harder tasks i.e. a large gap between average and worst-case performance, the algorithms are able to improve significantly over the baselines. For example, we observe a 5% improvement in performance in the case of CIFAR-100.

2. The *plug-and-play* framework allows for several different $W_n$ to enhance various *responsible AI* qualities at once. We demonstrate this with the partial domain aware setting (PDA), where the performance lead widens, indicating that RAI is able to jointly optimize effectively for both known and unknown subpopulations while Online GDRO suffers from some of the group information being unknown. In practice, one can construct many more novel sets $W_n$.

3. Although bigger (complex) models exhibit stronger performance than RAI ensembles, there are several caveats to this observation. Firstly, these models are ∼10-15 times larger than our base models. This limits their use w.r.t both training & inference compute required. However, RAI ensembles utilize a small number of much smaller models which can be individually trained quite easily. Even with these large models as base learners, constructing ensembles exhibits a performance boost, indicating that our framework is able to "boost" models of varying complexities.

# 8 Conclusion

Under the umbrella of "responsible AI", an emerging line of work has attempted to formalize desiderata ranging over ethics, fairness, robustness, and safety, among others. Many of these settings (Table 1) can be written as min-max problems involving optimizing some worst-case loss under a set of predefined distributions. For all the problems that can be framed as above, we introduce and study a general framework, which we refer to as Responsible AI (RAI) games. Our framework extends to classical boosting scenarios, offering boosting-based algorithms for RAI games alongside proven convergence guarantees. We propose practical algorithms to solve these games, as well as statistical analyses of solutions of these games. We find that RAI can guarantee multiple responsible AI aspects under appropriate choices of uncertainty sets.

## Acknowledgements

We acknowledge the support of DARPA via HR00112020006, and NSF via IIS-1909816.

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

# A    Broader Impact

Responsible AI has become an important topic as ML/AI systems increase in scale, and are being deployed in a variety of scenarios. Models not optimized for *responsible facets* can have disastrous consequences [Kalra and Paddock, 2016, Angwin et al., 2016, Fuster et al., 2018].

Our research promotes the principles of Responsible AI (RAI), encompassing ethics, fairness, and safety considerations. By providing a framework to consider these simultaneously, this research could help prevent scenarios where optimizing for one aspect unintentionally compromises another [Ma et al., 2022], mitigating the risk of creating biases or vulnerabilities in AI systems.

We also address the fragmentation in recent work around Responsible AI. The 'plug-and-play' nature of our approach allows for the easy adaptation of algorithms for different Responsible AI considerations. This adaptability could lead to more practical and user-friendly tools for implementing RAI across different applications.

Numerous ML models like Large language models, (GPT-3, GPT-4), are rapidly transforming numerous domains, including natural language processing, data analysis, content creation, and more. Given their remarkable ability to generate human-like text, they can be used to construct narratives, answer queries, or even automate customer service. However, with such capabilities come significant responsibilities, as these models can inadvertently perpetuate biases, misinformation, or harmful content if not correctly regulated. Therefore, ensuring the responsible behavior of these models is crucial [Chan, 2023, Bender et al., 2021]. Our research presents a general framework that can be pivotal in the responsible design of such large language models.

# B    Limitations

We now identify some limitations of our current work, along with corresponding future directions.

- To more concretely establish the empirical superiority of our optimization techniques, more experiments involving large over-parametrized models need to be conducted.
- The proposed generalization bounds are not tight for all risks. A more careful analysis would be needed for such a generalization.
- Our framework only handles uncertainty sets that are supported on the training data. It'd be interesting to generalize our framework further to support other uncertainty sets based on Wasserstein divergences that are not necessarily supported on the training data.
- Finally, the presence of outliers can often destabilize training of large models[Zhai et al., 2021b]. However, our current framework assumes the training data is un-corrupted. In future, we aim to extend our framework to support corruptions in the training data.
- We note that our framework can also be extended to the problem of adversarial test-time robustness where there is an adversary corrupting the inputs sent to the model during inference. Let $\mathcal{A}(x)$ be the set of perturbations that the adversary can add to input $x$. The uncertainty set in this case contains distributions supported on $\{(x', y') : \exists (x, y) \in \widehat{P}_{\text{data}} \text{ such that } x' \in x + \mathcal{A}(x), y' = y\}$.

Our framework primarily covers RAI aspects which can written as minmax problems. Here we provide some other notions of RAI which our framework does not directly cover.

**Fairness:**    Various notions of fairness have been studied by the ML community. While our framework captures minimax group fairness, it doesn't capture other notions of group fairness such as Demographic Parity [Louizos et al., 2015], Equality of Odds, Equality of Opportunity [Hardt et al., 2016b]. Our framework doesn't capture individual fairness notions.

**Robustness:**    While our framework covers group robustness and certain forms of distributional robustness, it doesn't cover robustness to Wasserstein perturbations and other (un)structured distribution shifts often studied in the domain generalization community [Wang et al., 2018, Addepalli et al., 2022, Cha et al., 2022].

# C  Terminology and Notation

## C.1  Terminology

**Strongly Convex Sets**  A set $A$ is $\lambda$-strongly convex w.r.t a norm $\|\cdot\|$, if, for any $x, y \in A, \gamma \in [0, 1]$, the $\|\cdot\|$ norm ball with origin $\gamma x + (1 - \gamma)y$ and radius $\gamma(1 - \gamma)\lambda\|x - y\|^2/2$ lies in $A$.

**$f$-divergence**  For any two probability distributions $P, Q$, $f$-divergence between $P$ and $Q$ is defined as $D_f(Q||P) = \mathbb{E}_P[f(dQ/dP)]$. Here $f : \mathbb{R}_+ \to \mathbb{R}$ is a convex function such that $f(1) = 0$.

## C.2  Notation

Table 3: Notation

| Symbol | Description |
|---|---|
| $X$ | Input Random Variable |
| $\mathcal{X}$ | Input Domain |
| $Y$ | Output Random Variable |
| $\mathcal{Y}$ | Output Domain |
| $Z$ | Sample Random Variable $(X, Y)$ |
| $\mathcal{Z}$ | Sample Domain $\mathcal{X} \times \mathcal{Y}$ |
| $S$ | Sample Set |
| $P_{data}$ | Data Generating Distribution |
| $\widehat{P}_{data}$ | Empirical Distribution (Uniform) over $S$ |
| $H$ | Set of Hypothesis |
| $Q$ | Distribution over Hypothesis from $H$ |
| $h$ | Any given hypothesis |
| $h_{rand;Q}$ | Randomized Ensemble given by $Q$ |
| $h_{det;Q}$ | De-randomized/Deterministic Classifier corresponding to $h_{rand;Q}$ |
| $l$ | Loss Function |
| $\widehat{R}(h)$ | Empirical Risk of $h$ |
| $W_n$ | Set of allowed sample weights (aka Uncertainty Set) |
| $\widehat{R}_{W_n}(h)$ | Empirical RAI Risk of $h$ |
| $R_W(h)$ | Population RAI Risk of $h$ |
| $\widehat{R}_{rand;W_n}(Q)$ | Randomized Ensemble RAI Risk |
| $KL(p||q)$ | KL-divergence Metric between $p$ and $q$ |
| $G_i$ | Subpopulation/Domain $i$ |
| Reg | Any given regularizer function |

# D  Background

## D.1  Two Player Zero-sum Games

Consider the following game between two players. One so-called "row player" playing actions $h \in H$, and the other "column player" playing actions $z \in Z$. Suppose that when the two players play actions $h, z$ respectively, the row player incurs a loss of $l(h, z) \in \mathbb{R}$, while the column player incurs a loss of $-l(h, z)$. The sum of the losses for the two players can be seen to be equal to zero so such a game is known as a two-player zero-sum game. It is common in such settings to refer to the gain $l(h, z)$ of the column player, rather than its loss of $-l(h, z)$. Both players try to maximize their gain/minimize their loss.

It is common in game theory to consider a linearized game in the space of probability measures, which is in general better behaved. To set up some notation, for any probability distributions $P_h$ over $H$, and $P_z$ over $Z$, define:

$$l(P_h, P_z) = \mathbb{E}_{P_h, P_z} l(h, z)$$

**Nash Equilibrium** A Nash Equilibrium (NE) is a stable state of a game where no player can gain by unilaterally changing their strategy while the other players keep theirs unchanged. In a two-player zero-sum game, a Nash Equilibrium is a pair of mixed strategies $(h^*, z^*)$ satisfying

$$\sup_{z \in Z} l(h^*, z) \leq l(h^*, z^*) \leq \inf_{h \in H} l(h, z^*)$$

Note that whenever a pure strategy NE exists, the minimax and maximin values of the game are equal to each other:

$$\inf_{h \in H} \sup_{z \in Z} l(h, z) = l(h^*, z^*) = \sup_{z \in Z} \inf_{h \in H} l(h, z)$$

What often exists is a mixed strategy NE, which is precisely a pure strategy NE of the linearized game. That is, $(P_h^*, P_z^*)$ is called a mixed strategy NE of the zero-sum game, if

$$\sup_{P_z \in \mathcal{P}_Z} l(P_h^*, P_z) \leq l(P_h^*, P_z^*) \leq \inf_{P_h \in \mathcal{P}_H} l(P_h, P_z^*)$$

For this paper, $Q \equiv P_h$, $w \equiv P_z$, $\Delta_H \equiv \mathcal{P}_H$ and $W_n \equiv \mathcal{P}_Z$.

**$\epsilon$-Approximate Nash Equilibrium** An $\epsilon$-approximate Nash Equilibrium is a relaxation of the Nash Equilibrium, where each player's strategy may not be the best response but is still within $\epsilon$ of the best response. Formally, a pair of mixed strategies $(P_h, P_z)$ is an $\epsilon$-approximate Nash Equilibrium if

$$\inf_{P_h \in \mathcal{P}_H} l(P_h, P_z) + \epsilon \geq l(P_h, P_z) \geq \sup_{P_z \in \mathcal{P}_Z} l(P_h, P_z) - \epsilon \tag{7}$$

**No Regret Algorithms** No-regret algorithms are a class of online algorithms used in repeated games. The regret of a player is defined as the difference between their cumulative payoff and the best cumulative payoff they could have achieved by consistently playing a single strategy. A no-regret algorithm guarantees that the average regret of a player goes to zero as the number of iterations (or rounds) tends to infinity. In the context of two-player zero-sum games, if both players follow no-regret algorithms, their average strategy profiles converge to the set of Nash Equilibria.

### D.2 Online Learning

A popular and widely used approach for solving min-max games is to rely on online learning algorithms [Hazan, 2016, Cesa-Bianchi and Lugosi, 2006]. In this approach, the row (minimization) player and the column (maximization) player play a repeated game against each other. Both players rely on online learning algorithms to choose their actions in each round of the game, with the objective of minimizing their respective regret. The following proposition shows that this repeated gameplay converges to a NE.

**Proposition 15** *([Gupta et al., 2020]) Consider a repeated game between the minimization and maximization players in the linearized game. Let $(P_h^t, P_z^t)$ be the actions chosen by the players in iteration $t$. Suppose the actions are such that the regret of each player satisfies:*

$$\sum_{t=1}^T l(P_h^t, P_z^t) - \inf_{h \in H} \sum_{t=1}^T l(h, P_z^t) \leq \epsilon_1(T)$$

$$\sup_{z \in Z} \sum_{t=1}^T l(P_h^t, z) - \sum_{t=1}^T l(P_h^t, P_z^t) \leq \epsilon_2(T)$$

*Let $P_{hAVG}, P_{zAVG}$ denote the mixture distributions $\frac{1}{T} \sum_{i=1}^T P_h^i$ and $\frac{1}{T} \sum_{i=1}^T P_z^i$. Then $(P_{hAVG}, P_{zAVG})$ is an $\epsilon$-approximate mixed NE of the game with:*

$$\epsilon = \frac{\epsilon_1(T) + \epsilon_2(T)}{T}$$

There exist several algorithms such as FTRL, FTPL, and Best Response (BR), which guarantee sub-linear regret. It is important to choose these algorithms appropriately, given the domains $H, Z$ as our choices impact the rate of convergence to a NE and also the computational complexity of the resulting algorithm.

### D.3 General Minimax Theorems (Proof of Proposition 1)

We first state the following convenient generalization of the original Von Neumann's minimax theorem.

**Proposition 16** *(Von Neumann-Fan minimax theorem, [Borwein, 2016]) Let $X$ and $Y$ be Banach spaces. Let $C \subset X$ be nonempty and convex, and let $D \subset Y$ be nonempty, weakly compact, and convex. Let $g : X \times Y \to R$ be convex with respect to $x \in C$ and concave and upper-semicontinuous with respect to $y \in D$, and weakly continuous in $y$ when restricted to $D$. Then,*

$$\sup_{y \in D} \inf_{x \in C} g(x, y) = \inf_{x \in C} \sup_{y \in D} g(x, y)$$

We now proceed to the proof of Proposition 1. Observe that a convex, compact $W_n$ satisfies the conditions for $D$ in the above proposition. Moreover, we have $C = \Delta_H$ i.e. the set of probability measures on $\Theta$. It is indeed nonempty and convex. Also, our $g$ is bilinear in $Q$ and $w$, and thus is convex-concave. Thus, Proposition 1 directly follows from the above result.

**Relaxations**    We can relax the assumption that $h$ is parameterized by a finite dimensional vector $\theta$. For simpler $H$, the minimax result directly holds with mild assumptions.

- If $H = \{h_1, h_2, ...h_n\}$ i.e. $H$ is finite. Then the original minimax theorem by Neumann holds for arbitrary functions $l$.

- If $H = \{h_1, h_1, ...\}$ i.e. $H$ is denumerable. We further assume $l$ is a bounded loss function. Then from Theorem 3.1 from [Wald, 1945] to compact and convex $W$ over $n$ (finite) datapoints, we can conclude the relation holds.

Hoever, minimax theorems for more general $H$ require other conditions like the continuity of loss, compactness in function space, etc. See [Simons, 1995] and [Raghavan, 1994].

# E    Ensemble RAI Games

## E.1    Alternative Definitions: Deterministic Ensembles

Alternative definitions for deterministic ensembles could be considered. For example, one could consider $h_{\text{det};Q}(x) = \arg\min_{y \in \mathcal{Y}} \mathbb{E}_{h \sim Q} \ell(h(x), y)$. [Cotter et al., 2019, Wu et al., 2022] designed other more sophisticated strategies, but these are largely domain dependent. For reasons that will be explained later, we stick with Definition 3 in this work. For regression, a popular de-randomization strategy is to compute the expected prediction: $h_{\text{det};Q}(x) = \mathbb{E}_{h \sim Q}[h(x)]$.

## E.2    Proofs

### E.2.1    Proposition 2

*Proof.*

$$\sup_{w \in \Delta_n} \widehat{\mathbb{E}}_w \mathbb{I}[h_{\text{det};Q}(x) \neq y] = \sup_{i \in [n]} \mathbb{I}[y_i \neq \arg\max_{y \in \mathcal{Y}} \mathbb{E}_Q[h(x_i) = y]]$$
$$= \mathbb{I}[\sup_{w \in \Delta_n} \mathbb{E}_w \mathbb{E}_Q \mathbb{I}[h(x) \neq y] \geq 1/2]$$
$$= \mathbb{I}[\widehat{R}_{W_n}(h_{\text{rand};Q}) \geq 1/2]$$

as required.                                                                                            □

### E.2.2 Proposition 3

*Proof.* Denote $y_Q(x) = \arg\max_{y \in \mathcal{Y}} \mathbb{E}_Q(h(x) = y)$. Then,

$$
\begin{aligned}
\widehat{R}_{W_n}(h_{\text{det};Q}) &= \sup_{w \in W_n} \mathbb{E}_w \ell(y_Q(x), y) \\
&\leq \sup_{w \in W_n} \mathbb{E}_w \ell(y_Q(x), y) \frac{P_Q(h(x) = y_Q(x))}{1/\gamma_Q} \\
&\leq \gamma_Q \sup_{w \in W_n} \mathbb{E}_w \sum_{y' \in \mathcal{Y}} \ell(y', y) P_Q(h(x) = y') \\
&= \gamma_Q \sup_{w \in W_n} \mathbb{E}_w \mathbb{E}_Q \sum_{y' \in \mathcal{Y}} \ell(y', y) \mathbb{I}[h(x) = y'] \\
&= \gamma_Q \sup_{w \in W_n} \mathbb{E}_w \mathbb{E}_Q \ell(h(x), y) \\
&= \gamma_Q \widehat{R}_{W_n}(h_{\text{rand};Q}),
\end{aligned}
$$

as required. $\qquad\qquad\qquad\qquad\qquad\qquad\qquad\qquad\qquad\qquad\qquad\qquad\square$

## F  Algorithms

### F.1  Convergence Rates

#### F.1.1  Gameplay

We begin with the following lemma adapted from [McMahan, 2017] (Theorem-1)

**Lemma 17** *Consider the setting of Algorithm 1, and further assume that $\eta^t \geq \eta^{t-1} > 0$, $Reg(w) \geq 0$, $\eta^t Reg(w)$ is 1-strongly concave w.r.t. some norm $\|.\|_{(t)}$. Then for any $w^* \in W_n$ and any $T > 0$, we have:*

$$
Regret_{D,T} \leq \eta^{T-1} Reg(w^*) + \frac{1}{2} \sum_{t=1}^{T} \|l^t\|_{(t-1),*}^2 \quad (\text{where } l_i^t = l(h^t(x_i), y_i))
$$

Consider $\eta^t = \eta$ and $\|.\|_{(t)} = \sqrt{\eta}\|.\|_1$, then

$$
Regret_{D,T} \leq \eta \mathrm{Reg}(w^*) + \frac{1}{2\eta} \sum_{t=1}^{T} \|l^t\|_*^2 \leq \eta D_R^2 + \frac{T}{2\eta}
$$

Moreover, as H-player plays BR,

$$
Regret_{H,T} \leq 0
$$

Using Proposition 15, we achieve $\epsilon$-approximate NE with:

$$
\epsilon = \epsilon_T \leq \frac{Regret_{H,T} + Regret_{D,T}}{T} = \frac{\eta D_R^2}{T} + \mathcal{O}\left(\frac{1}{\eta}\right)
$$

Using definition in Equation 7 gives us the required result.

#### F.1.2  Greedy

**FW Update**  Note that we are trying to minimize the objective $L_\eta(Q)$ w.r.t $Q$ by the FW update. Using properties of Fenchel conjugates, it is well known that $L_\eta(Q)$ is $\frac{1}{\eta}$ smooth w.r.t. $\|.\|_1$. Also, the diameter of the simplex $\Delta_H$ w.r.t. $\|.\|_1$ is $\leq 1$. By [Jaggi, 2013] (Lemma 7), we have $C_f \leq \frac{1}{\eta}$, and thus by [Jaggi, 2013] Theorem 1, we have:

$$
L_\eta(Q^T) - \min_Q L_\eta(Q) \leq \frac{2}{\eta(T+2)} = \mathcal{O}\left(\frac{1}{\eta T}\right)
$$

Using the definition of $L_\eta(Q)$,

$$
L(Q^T) - \min_Q L(Q) \leq \eta D_R^2 + \mathcal{O}\left(\frac{1}{\eta T}\right)
$$

**Gen-AdaBoost Update** This update boils down to a standard coordinate descent update on convex and $\frac{1}{\eta}$ smooth $L_\eta(Q)$ (w.r.t $\|.\|_1$). Following along the lines of analysis in [Boyd and Vandenberghe, 2004] (Section 9.4.3),

$$L_\eta(Q^t) \leq L_\eta(Q^{t-1}) - \frac{\eta}{2}\|\nabla_Q L_\eta(Q^{t-1})\|_\infty^2$$

Using this decent equation, we can follow standard gradient descent analysis to get:

$$L_\eta(Q^T) - \min_Q L_\eta(Q) \leq \mathcal{O}\left(\frac{1}{\eta T}\right)$$

The rest of the argument will go through as above.

### F.2 Closed Form Updates for different uncertainty sets

In this section, we derive closed-form updates for Equation 2 for the entropic regularizer (also mentioned below). We consider common settings mentioned in Table 1.

$$w^t \leftarrow \operatorname*{argmax}_{w \in W_n} \sum_{s=1}^{t-1} \mathbb{E}_w \ell(h^s(x), y) - \eta^{t-1} \sum w \log(w)$$

- $W_n = \{\widehat{P}_{data}\}$ **(Empirical Risk Minimization)**

$$w^t \leftarrow \widehat{P}_{data}$$

- $W_n = \Delta_n$ **(Worst Case Margin)**

$$w^t \leftarrow \frac{u^t}{\|u^t\|_1} \quad \text{where} \quad u_i^t \leftarrow \exp\left(-\frac{\sum_{s=1}^{t-1} l(h^s(x_i), y_i)}{\eta^{t-1}}\right)$$

- $W_n = \{w : w \in \Delta_n, w \preceq \frac{1}{\alpha n}\}$ **($\alpha$-CVaR)**

$$w_i^t \leftarrow \min\left(\frac{1}{\alpha n}, \exp\left(-\frac{\sum_{s=1}^{t-1} l(h^s(x_i), y_i)}{\eta^{t-1}} - \lambda\right)\right) \quad \text{for } \lambda \quad \text{S.T.} \sum_i w_i^t = 1$$

Algorithm 3 describes a projection procedure to find such $\lambda$ in $\mathcal{O}(n \log n)$ time.

---

**Algorithm 3** Projection for $\alpha$-CVaR set

---

**Input:** $l, \eta, \alpha$
1: $y_i \leftarrow \exp\left(-\frac{\sum_{s=1}^{t-1} l(h^s(x_i), y_i)}{\eta^{t-1}}\right)$
2: $v \leftarrow \frac{1}{\alpha n}$
3: **if** $\frac{y_i}{\|y_i\|_1} \leq v \ \forall i$ **then**
4: $\quad w_i \leftarrow \frac{y_i}{\|y_i\|_1}$
5: $\quad$ **return** $w$
6: **else**
7: $\quad y_{(i)} \leftarrow \text{sort}\{y_i\} \quad$ S.T. $\quad y_{(i)} \geq y_{(j)} \ \forall \ i \leq j$
8: $\quad$ **function** CANDIDATE(m)
9: $\quad\quad c_m \leftarrow \frac{v}{y_{(m)}}$
10: $\quad\quad S_m \leftarrow \sum_i v\mathbb{1}_{c_m y_{(i)} \geq v} + c_m y_{(i)}\mathbb{1}_{c_m y_{(i)} < v}$
11: $\quad\quad$ **return** $S_m$
12: $\quad$ **end function**
13: $\quad$ Let $m^* \leftarrow$ binary search for the smallest $m$ such that CANDIDATE$(m) \leq 1$
14: $\quad c_{m^*} \leftarrow \frac{1 - \sum_{i \leq m^*} v}{\sum_{i > m^*} y_{(i)}}$
15: $\quad w_i \leftarrow \min(c_{m^*} y_i, v)$
16: $\quad$ **return** $w$
17: **end if**

---

- $W_n = \{w : D(w\|\widehat{P}_{\textbf{data}}) \leq \rho_n\}$ **(DRO)** For general $f$-divergences, there do not exist closed form updates for $w^t$. However, they can still be empirically solved using FW-like updates.

- $W_n = \{\widehat{P}_{\textbf{data}}(G_1), \widehat{P}_{\textbf{data}}(G_2), \dots \widehat{P}_{\textbf{data}}(G_K)\}$ **(Group DRO)**

$$w^t \leftarrow \frac{u^t}{\|u^t\|_1} \quad \text{where} \quad u_i^t \leftarrow \exp\left(-\frac{\sum_{s=1}^{t-1}\sum_{i \in G_k} l(h^s(x_i), y_i)}{\eta^{t-1}s_k}\right) \text{ for } i \in G_k, s_k = |G_k|$$

### F.3   Proof of Corollary 8

*Proof.* Note that Reg is 1-strongly concave w.r.t $\|.\|_1$ and $\|.\|_2$. A conservative upper bound for $D_R^2 \leq \log(n)$ for all $W_n(\subseteq \Delta_n)$. Thus, we can thus take appropriate values of $\eta$ to get:

$$L(Q^T) \leq \min_Q L(Q) + \mathcal{O}\left(\sqrt{\frac{\log(n)}{T}}\right)$$

Thus, we have $\epsilon \sim \mathcal{O}\left(\sqrt{\frac{\log(n)}{T}}\right)$ from Proposition 15. $\qquad\square$

## G   Generalization

### G.1   Population Risk

We first present a proposition which gives an equivalent characterization of the RAI population risk

**Proposition 18** *The following are equivalent characterizations of the population RAI risk*

*1.*

$$R_W(h) = \sup_{P: P \ll P_{data}, \frac{dP}{dP_{data}} \in W} \mathbb{E}_P[\ell(h(x), y)].$$

*2.*

$$R_W(h) = \sup_{r: \mathcal{Z} \mapsto \mathbb{R}_+, \int r(z)dP_{data}(z)=1, r \in W} \mathbb{E}_{P_{data}}[r(z)\ell_h(z)].$$

*Proof.* The equivalence between $(1)$ and $(2)$ follows by reparameterizing $P$ in $(1)$ as follows

$$dP(z) = r(z)dP_{\text{data}}(z),$$

for some $r(z) \geq 0$ $\qquad\square$

Now suppose the uncertainty set $W$ is as defined in Equation (5). The next proposition uses duality to derive an equivalent characterization of the RAI risk in this setting.

**Proposition 19** *Suppose the uncertainty set $W$ is as specified in Equation* (5)*. Then for any hypothesis $h$, the population RAI risk can be equivalently written as*

$$R_W(h) = \inf_{\lambda \geq \mathbf{0}, \tau} \mathbb{E}_{P_{data}} G_\lambda^*(\ell_h(z) - \tau) + \sum_{i=1}^m \lambda_i c_i + \tau, \tag{8}$$

*where $G_\lambda^*$ is the Fenchel conjugate of $G_\lambda(t) = \sum_{i=1}^m \lambda_i g_i(t)$.*

*Proof.* We rely on duality to prove the proposition. First observe that the population RAI risk can be rewritten as

$$R_W(h) = \sup_{r: \mathcal{Z} \mapsto \mathbb{R}_+, \int r(z)dP_{\text{data}}(z)=1, r \in W} \mathbb{E}_{P_{\text{data}}}[r(z)\ell_h(z)]$$

$$\overset{(a)}{=} \sup_{r: \mathcal{Z} \mapsto \mathbb{R}_+} \inf_{\lambda \geq \mathbf{0}, \tau} \mathbb{E}_{P_{\text{data}}}[r(z)\ell_h(z)] + \tau\left(1 - \int r(z)dP_{\text{data}}(z)\right) + \sum_{i=1}^m \lambda_i\left(c_i - \int g_i(r(z))dP_{\text{data}}(z)\right)$$

Since the above objective is concave in $r$ and linear in $\lambda, \tau$, we rely on Lagrangian duality to rewrite it as

$$R_W(h) = \inf_{\lambda \geq \mathbf{0}, \tau} \sup_{r: \mathcal{Z} \mapsto \mathbb{R}_+} L(r, \lambda, \tau),$$

where $L(r, \lambda, \tau)$ is defined as:

$$L(r, \lambda, \tau) = \mathbb{E}_{P_{\text{data}}} \left[ r(z)(\ell_h(z) - \tau) - \sum_{i=1}^{m} \lambda_i g_i(r(z)) \right] + \sum_{i=1}^{m} \lambda_i c_i + \tau.$$

Recall the interchangeability theorem:

$$\inf_{r \in \mathcal{H}} \int F(r(z), z) p(z) dz = \int \left( \inf_{t \in \mathbb{R}} F(t, z) \right) p(z) dz,$$

so long as the space $\mathcal{H}$ is decomposable. Since in our case we are working with the set $L_1(\mathcal{Z}, P) = \{r : \mathcal{Z} \mapsto \mathbb{R} : \int r(z) p(z) dz = 1\}$, which is decomposable, we can apply the interchangeability theorem to get:

$$\sup_{r: \mathcal{Z} \mapsto \mathbb{R}_+} \mathbb{E}_{P_{\text{data}}} \left[ r(z)(\ell_h(z) - \tau) - \sum_{i=1}^{m} \lambda_i g_i(r(z)) \right] = \mathbb{E}_{P_{\text{data}}} \sup_{t \geq 0} \left[ t(\ell_h(z) - \tau) - \sum_{i=1}^{m} \lambda_i g_i(t) \right]$$

$$= \mathbb{E}_{P_{\text{data}}} G_\lambda^*(\ell_h(z) - \tau),$$

where $G_\lambda(t) = \sum_{i=1}^{m} \lambda_i g_i(t)$, and $G_\lambda^*$ is its Fenchel conjugate. so that:

$$R(\ell_h) = \inf_{\lambda \geq \mathbf{0}, \tau} \sum_{i=1}^{m} \lambda_i c_i + \tau + \mathbb{E}_{P_{\text{data}}} G_\lambda^*(\ell_h(z) - \tau).$$

$\square$

We have the following properties of the Fenchel conjugate $G_\lambda^*(t)$. These follow from the properties of Fenchel conjugates described in Rockafellar [1970].

**Lemma 20** *Consider the setting of Proposition 19. The Fenchel conjugate $G_\lambda^*$ is convex, differentiable and an increasing function that satisfies*

$$\frac{dG_\lambda^*(x)}{dx} \geq 0, \ \forall x \in \mathbb{R}.$$

**Proof of Proposition 9** For the sake of clarity, we first state Proposition 9 below.

**Proposition 21 (Bayes optimal classifier)** *Consider the problem of binary classification where $\mathcal{Y} = \{-1, +1\}$. Suppose $\ell(h(x), y) = \phi(yh(x))$ for some $\phi : \mathbb{R} \to [0, \infty)$ which is either the 0/1 loss, or a convex loss function that is differentiable at 0 with $\phi'(0) < 0$. Suppose the uncertainty set $W$ is as specified in Equation (5). Moreover, suppose $\{g_i\}_{i=1\ldots m}$ are convex and differentiable functions. Then, the vanilla Bayes optimal classifier is also a RAI-Bayes optimal classifier.*

*Proof.* Following Proposition 19 it is easy to see that the RAI Bayes optimal classifier is the minimizer of the following problem

$$\inf_h \inf_{\lambda \geq \mathbf{0}, \tau} \sum_{i=1}^{m} \lambda_i c_i + \tau + \mathbb{E}_{P_{\text{data}}} G_\lambda^*(\phi(yh(x)) - \tau),$$

where the minimization over $h$ is over the set of all classifiers. For any fixed $(\lambda, \tau)$, we now show that the classifier $h$ that minimizes the above optimization problem is a vanilla Bayes optimal classifier. First note that the above optimization problem, for a fixed $(\lambda, \tau)$, can be rewritten as

$$\inf_h \mathbb{E}_{P_{\text{data}}} G_\lambda^*(\phi(yh(x)) - \tau).$$

Using the interchangeability theorem, we can further rewrite this as

$$\mathbb{E}_{P_{\text{data}}^x} \left[ \inf_{u \in \{-1, +1\}} \mathbb{E}_{P_{\text{data}}(\cdot|x)} \left[ G_\lambda^*(\phi(uy) - \tau) \right] \Big| x \right].$$

Here, $P_{\text{data}}^x$ is the marginal distribution of $P_{\text{data}}$ over $x$, and $P_{\text{data}}(\cdot|x)$ is the distribution of $y$ conditioned on $x$.

Now suppose $\phi$ is the 0/1 loss

$$\phi(x) = \begin{cases} 0, & \text{if } x > 0 \\ 1, & \text{otherwise} \end{cases}.$$

Recall, $G_\lambda^*$ is an increasing function (see Lemma 20). So $G_\lambda^*(-\tau) \leq G_\lambda^*(1-\tau)$. Using this, it is easy to see that for any $x$, the following is a minimizer of $\inf_{u \in \{-1,1\}} \mathbb{E}_{P_{\text{data}}(\cdot|x)} [G_\lambda^*(\phi(uy) - \tau)]$

$$u^* = \begin{cases} 1, & \text{if } P_{\text{data}}(y = 1|x) \geq \frac{1}{2} \\ -1, & \text{otherwise} \end{cases}.$$

This shows that the vanilla Bayes optimal classifier is a minimizer of the population RAI risk.

Now suppose $\phi : \mathbb{R} \to [0, \infty)$ is convex, differentiable at 0 with $\phi'(0) < 0$. Moreover, suppose $h : \mathcal{X} \to \mathbb{R}$ is a real valued classifier. In this case, the RAI Bayes optimal classifier is a minimizer of the following objective

$$\mathbb{E}_{P_{\text{data}}^x} \left[ \inf_{u \in \mathbb{R}} \mathbb{E}_{P_{\text{data}}(\cdot|x)} [G_\lambda^*(\phi(uy) - \tau)] \,\Big|\, x \right].$$

Let $\iota(x) = G_\lambda^*(\phi(x) - \tau)$. It is easy to see that $\iota(x)$ is convex. This is because $\iota'(x) = (G_\lambda^*)'(\phi(x) - \tau)\phi'(x)$ is an increasing function; this follows from the fact that $G_\lambda^*$ is convex with non-negative gradients. Moreover, $\iota'(0) = (G_\lambda^*)'(\phi(0) - \tau)\phi'(0) \leq 0$. Then, Bartlett et al. [2006], Tewari and Bartlett [2007] show that for any $x$, the following $u^*$ is a minimizer of the inner optimizatin problem: $u^* > 0$ if $P_{\text{data}}(y = 1|x) \geq \frac{1}{2}$, $u^* < 0$ otherwise. This shows that vanilla Bayes optimal classifier is minimizer of the population RAI risk. $\qquad\square$

## G.2  Generalization Guarantees

### G.2.1  Proof of Proposition 11

Proposition 11 directly follows from Proposition 19.

### G.2.2  Proof of Theorem 12

We first present a key concentration result we use in the proof.

**Lemma 22 (Hoeffding bound [Wainwright, 2019])** *Suppose that the random variables $\{X_i\}_{i=1}^n$ are independent with mean $\mu_i$, and bounded between $[a, b]$. Then for any $t \geq 0$, we have*

$$\mathbb{P}\left( |\sum_{i=1}^n X_i - \mu_i| \geq t \right) \leq 2 \exp\left( -\frac{2t^2}{n(b-a)^2} \right).$$

We now proceed to the proof of the Theorem. Following Proposition 11, we know that the population and empirical RAI risk of a classifier $h$ can be written as

$$R_W(h) = \inf_{\lambda \geq \mathbf{0}, \tau} \sum_{i=1}^m \lambda_i c_i + \tau + \mathbb{E}_{P_{\text{data}}} G_\lambda^*(\ell_h(z) - \tau)$$

$$\widehat{R}_{W_n}(h) = \inf_{\lambda \geq \mathbf{0}, \tau} \sum_{i=1}^m \lambda_i c_i + \tau + \mathbb{E}_{\widehat{P}_{\text{data}}} G_\lambda^*(\ell_h(z) - \tau)$$

Our goal here is to bound the following quantity for any given $h$:

$$|R_W(h) - \widehat{R}_{W_n}(h)| \leq \sup_{(\lambda, \tau) \in \mathcal{E}, \lambda : \min_i \lambda_i \geq \underline{\Lambda}} \left| \mathbb{E}_{P_{\text{data}}} G_\lambda^*(\ell_h(z) - \tau) - \mathbb{E}_{\widehat{P}_{\text{data}}} G_\lambda^*(\ell_h(z) - \tau) \right|.$$

The rest of the proof focuses on bounding the RHS of the above equation. The overall idea is to first provide a high probability bound of the RHS for any given $\lambda, \tau$. Next, we take a union bound over all feasible $(\lambda, \tau)$'s by constructing an appropriate $\epsilon$-net.

**Fixed $\lambda, \tau$.** Observe that $G_\lambda^*(\ell_h(z) - \tau)$ is bounded and satisfies

$$G_\lambda^*(-\tau) \le G_\lambda^*(\ell_h(z) - \tau) \le G_\lambda^*(B - \tau).$$

This follows from the fact that $G_\lambda^*$ is an increasing function (see Proposition 20), and $\ell_h$ is bounded between $0$ and $B$. From Heoffding bound we know that for any fixed $h \in H$, the following holds with probability at least $1 - 2e^{-t}$

$$\left| \mathbb{E}_{P_{\text{data}}} G_\lambda^*(\ell_h(z) - \tau) - \mathbb{E}_{\widehat{P}_{\text{data}}} G_\lambda^*(\ell_h(z) - \tau) \right| \le G\sqrt{\frac{t}{n}}.$$

**Union bound over $\lambda, \tau$.** Define set $\mathcal{E}'$ as

$$\mathcal{E}' := \{(\lambda, \tau) : \lambda \ge 0, \min_i \lambda_i \ge \Lambda\} \cap \mathcal{E}. \tag{9}$$

Let $N(\mathcal{E}', \epsilon, \|\cdot\|_2)$ be the $\epsilon$-net over $\mathcal{E}'$ in $\|\cdot\|_2$ norm. It is well known that there exists such a set whose size is upper bounded by [Wainwright, 2019]

$$|N(\mathcal{E}', \epsilon, \|\cdot\|_2)| \le O\left(\frac{\bar{\Lambda} + T}{\epsilon}\right)^{m+1}$$

. For any $(\lambda, \tau) \in \mathcal{E}'$, let $(\lambda_\epsilon, \tau_\epsilon)$ be an element in $N(\mathcal{E}', \epsilon, \|\cdot\|_2)$ that is $\epsilon$-close to $(\lambda, \tau)$. Now consider the following

$$\sup_{(\lambda,\tau)\in\mathcal{E}'} \left| \mathbb{E}_{P_{\text{data}}} G_\lambda^*(\ell_h(z) - \tau) - \mathbb{E}_{\widehat{P}_{\text{data}}} G_\lambda^*(\ell_h(z) - \tau) \right|$$

$$\le \sup_{(\lambda,\tau)\in N(\mathcal{E}',\epsilon,\|\cdot\|_2)} \left| \mathbb{E}_{P_{\text{data}}} G_\lambda^*(\ell_h(z) - \tau) - \mathbb{E}_{\widehat{P}_{\text{data}}} G_\lambda^*(\ell_h(z) - \tau) \right|$$

$$+ \sup_{(\lambda,\tau)\in\mathcal{E}'} \left| \mathbb{E}_{P_{\text{data}}} G_\lambda^*(\ell_h(z) - \tau) - \mathbb{E}_{P_{\text{data}}} G_{\lambda_\epsilon}^*(\ell_h(z) - \tau_\epsilon) \right|$$

$$+ \sup_{(\lambda,\tau)\in\mathcal{E}'} \left| \mathbb{E}_{\widehat{P}_{\text{data}}} G_\lambda^*(\ell_h(z) - \tau) - \mathbb{E}_{\widehat{P}_{\text{data}}} G_{\lambda_\epsilon}^*(\ell_h(z) - \tau_\epsilon) \right|$$

Since $G_\lambda^*$ is $L$-Lipschitz, the last two terms in the RHS above can be upper bounded by $L\epsilon$. Substituting this in the above equation, we get

$$\sup_{(\lambda,\tau)\in\mathcal{E}'} \left| \mathbb{E}_{P_{\text{data}}} G_\lambda^*(\ell_h(z) - \tau) - \mathbb{E}_{\widehat{P}_{\text{data}}} G_\lambda^*(\ell_h(z) - \tau) \right|$$

$$\le \sup_{(\lambda,\tau)\in N(\mathcal{E}',\epsilon,\|\cdot\|_2)} \left| \mathbb{E}_{P_{\text{data}}} G_\lambda^*(\ell_h(z) - \tau) - \mathbb{E}_{\widehat{P}_{\text{data}}} G_\lambda^*(\ell_h(z) - \tau) \right| + 2L\epsilon$$

$$\overset{(a)}{\le} G\sqrt{\frac{t}{n}} + 2L\epsilon,$$

where $(a)$ follows from Equation (9), and holds with probability at least $1 - \left(\frac{\bar{\Lambda}+T}{\epsilon}\right)^{m+1} e^{-t}$. Choosing $\epsilon = \frac{G}{L}\sqrt{\frac{t}{n}}$, we get the desired result.

### G.3 Proof of Corollary 13

The proof follows from a standard covering number argument. For any $h \in H$, let $h_\epsilon$ be the point in the $\epsilon$-net that is closest to $h$. Then we have

$$\sup_{h\in H} |R_W(h) - \widehat{R}_{W_n}(h)| \le \sup_{h\in N(H,\epsilon_n,\|\cdot\|_{L^\infty(\mathcal{Z})})} |R_W(h) - \widehat{R}_{W_n}(h)|$$

$$+ \sup_{h\in H} |R_W(h) - R_W(h_\epsilon)| + \sup_{h\in H} |\widehat{R}_{W_n}(h) - \widehat{R}_{W_n}(h_\epsilon)|$$

Observe that the last two terms above are bounded by $\epsilon_n$. Also observe that the first term in the RHS can be upper bounded by $10n^{-1/2}G(\sqrt{t + m\log(nL)})$ with probability at least $1 - 2N(H, \epsilon_n, \|\cdot\|_{L^\infty(\mathcal{Z})})e^{-t}$. Combining these two and substituting the value of $\epsilon_n$ gives us the required result.

## H   Algorithms: Further Discussion

### H.1   Equivalence Conditions for RAI Algorithms (1 and 2)

**Proposition 23** *Assume that we set $\alpha_t = \frac{1}{t}$ and perform coordinate descent update in Algorithm 2 (FW) i.e. $G^t = \operatorname{argmin}_{h \in H} \langle h, \nabla_Q L_\eta(Q^{t-1}) \rangle$, then the update is equivalent to the update given by Algorithm 1 with $\eta^t = \eta t$.*

*Proof.* From Equation 3, and using the fact that $L_\eta(Q)$ can be written as a fenchel conjugate, we know that

$$\nabla_Q L_\eta(Q^{t-1}) = \operatorname*{argmax}_{w \in W_n} \mathbb{E}_{h \sim Q^{t-1}} \mathbb{E}_w \ell(h(x), y) + \eta \mathrm{Reg}(w)$$

$$= \operatorname*{argmax}_{w \in W_n} \sum_{s=1}^{t-1} \mathbb{E}_w \ell(h^s(x), y) + \eta t \mathrm{Reg}(w) \quad \left( \text{as } \alpha^t = \frac{1}{t} \right)$$

This matches Equation 2 with $\eta^{t-1} = \eta t$. Moreover,

$$G^t = \operatorname*{argmin}_{h \in H} \mathbb{E}_{h \in H} \mathbb{E}_{w^t} l(h(x), y) \quad \text{where} \quad w^t = \nabla_Q L_\eta(Q^{t-1})$$

This corresponds to the update for $h^t$ in Algorithm 1. Thus, we have our equivalence. $\qquad \square$

### H.2   Weak Learning Conditions

For the well-known scenario of binary classification and zero-one loss, we recover the quasi-AdaBoost weak learning condition:

**Proposition 24** *Consider the scenario of binary classification and $l$ as the zero-one loss. If the H-player only plays an approximate best response strategy i.e. $h^t$ satisfies $\mathbb{E}_{w^t} \ell(h^t(x), y) \leq 1/2 - \gamma$ for some $\gamma > 0$, then $\widehat{R}_{W_n}(h_{det:Q^T}) = 0$ for $T > T_0$ for some large enough $T_0$.*

*Proof.* Since the D-player uses regret optimal strategy, we have that:

$$\frac{1}{T} \sum_{t=1}^{T} \mathbb{E}_{w^t} \ell(h^t(x), y) \geq \max_{w \in W_n} \frac{1}{T} \sum_{t=1}^{T} \mathbb{E}_w \ell(h^t(x), y) - \epsilon_T,$$

while from the approximate-BR condition we have that:

$$\frac{1}{T} \sum_{t=1}^{T} \mathbb{E}_{w^t} \ell(h^t(x), y) \leq 1/2 - \gamma,$$

so that we have:

$$\max_{w \in W_n} \frac{1}{T} \sum_{t=1}^{T} \mathbb{E}_w \ell(h^t(x), y) \leq 1/2 - \gamma + \epsilon_T,$$

so that for $T > T_0$ large enough so that $\epsilon_T < \gamma/2$, we have that:

$$\max_{w \in W_n} \frac{1}{T} \sum_{t=1}^{T} \mathbb{E}_w \ell(h^t(x), y) < 1/2 - \gamma/2.$$

As $Q^T$ assigns mass $1/T$ to each of $\{h^t\}_{t=1}^T$, we have:

$$\widehat{R}_{W_n}(h_{rand;Q^T}) = \max_{w \in W_n} \mathbb{E}_{Q^T} \mathbb{E}_w \ell(h^t(x), y) < 1/2 - \gamma/2$$

$$\implies \widehat{R}_{W_n}(h_{det:Q^T}) = 0 \quad \text{(from Proposition 2)}$$

Hence, we see that $h_{det;Q^T}$ incurs zero error. $\qquad \square$

For the general setting, we have a slightly *stronger* weak learning condition, which follows from the analysis of Frank-Wolfe update [Freund and Grigas, 2014, Jaggi, 2013].

**Proposition 25** *Consider Algorithm 2 with the FW update and Reg(.) as a 1-strongly concave regularize w.r.t.* $\|.\|_1$. *If* $G^t$ *satisfies* $G^t \leq \min_Q \left\langle Q, \nabla_Q L_\eta(Q^{t-1}) \right\rangle + \delta_t$, *where* $\{\delta_k\}$ *is the sequence of approximation errors with* $\delta_t \geq 0$, *then:*

1. *If* $\delta_t \leq \frac{\epsilon}{\eta(t+2)}$ *i.e. decaying errors, then* $L_\eta(Q^T) \leq \min_Q L_\eta(Q) + \frac{2(1+\epsilon)}{\eta(T+2)}$

2. *If* $\delta_t \leq \frac{\epsilon}{\eta}$ *i.e. constant errors, then* $L_\eta(Q^T) \leq \min_Q L_\eta(Q) + \frac{2}{\eta(T+2)} + \frac{\epsilon}{\eta}$

*Proof.* Note that we are trying to minimize the objective $L_\eta(Q)$ w.r.t $Q$ by the FW update. Using properties of Fenchel conjugates, it is well known that $L_\eta(Q)$ is $\frac{1}{\eta}$ smooth w.r.t. $\|.\|_1$. Also, the diameter of the simplex $\Delta_H$ w.r.t. $\|.\|_1$ is $\leq 1$.

1. By [Jaggi, 2013] (Lemma 7, Theorem 1), we have $C_f \leq \frac{1}{\eta}$, and thus, we have:

$$L_\eta(Q^T) - \min_Q L_\eta(Q) \leq \frac{2(1+\epsilon)}{\eta(T+2)}$$

2. By [Freund and Grigas, 2014] (Theorem 5.1), in case of approximation errors, the FW/optimality gap converges as before along with a convex combination of errors at each time step i.e. if we are able to solve the linear optimization problem within constant error $\frac{\epsilon}{\eta}$, then these errors do not accumulate. Moreover, the convex combination can be bound by the maximum error possible and we get,

$$L_\eta(Q^T) - \min_Q L_\eta(Q) \leq \frac{2}{\eta(T+2)} + \frac{\epsilon}{\eta}$$

$\square$

# I    Experiments

RAI games constitute an optimization paradigm that goes beyond traditional approaches such as distributionally robust optimization, fairness, and worst-case performance. We have seen that for specific uncertainty sets $W$, RAI Games optimize over well-established robust optimization objectives. As such, the purpose of our experiments is to demonstrate the practicality and generality of our proposed strategies, *rather* than establishing state-of-the-art over baselines. Given a large number of possible $W$, we do not attempt an exhaustive empirical analysis. Instead, we underscore the `plug-and-play` nature of RAI Games.

## I.1   Setup

**Subpopulation Shift**    A prevalent scenario in machine learning involves subpopulation shift, necessitating a model that performs effectively on the data distribution of each subpopulation (or domain). We explore the following variations of this setting:

- **Domain Oblivious (DO).** Recent work [Hashimoto et al., 2018], [Lahoti et al., 2020], [Zhai et al., 2021a] studies the domain-oblivious setting, where the training algorithm lacks knowledge of the domain definition. In this case, approaches like $\alpha$-CVaR and $\chi^2$-DRO aim to maximize performance over a general notion of the worst-off subpopulation.
- **Domain Aware (DA).** Several prior works [Sagawa et al., 2019] have investigated the domain-aware setting, in which all domain definitions and memberships are known during training.
- **Partially Domain-Aware (PDA).** More realistically, in real-world applications, there usually exist multiple domain definitions. Moreover, some of these domain definitions may be known during training, while others remain unknown. The model must then perform well on instances from all domains, regardless of whether their definition is known. This setting is challenging as it necessitates the model to learn both domain-invariant and domain-specific features and to generalize well to new instances from unknown domains.

## I.2 Further Details for Section 7

**Base Learners** We use linear classifiers, WRN-28-1, and WRN-28-5 [Zagoruyko and Komodakis, 2016] as base classifiers for COMPAS, CIFAR-10 and CIFAR-100 respectively. To get a sense of performance improvements, we also benchmark performance with larger models, namely a three-hidden-layer neural network for COMPAS and WRN-34-10 for CIFAR-10/100.

**Proposed Methods** This paper introduces two categories of algorithms. We elect not to present results for Algorithm 1, which we notice has similar performance to Algorithm 2. Conversely, we provide in-depth experimental analyses for both updates of Algorithm 2, which warrant some special attention due to due to their relation to AdaBoost. In this section, we refer to the FW and Gen-AdaBoost updates as RAI-FW and RAI-GA, respectively. Our implemented versions incorporate a few alterations: 1. We track the un-regularized objective value from Equation 1 for the validation set. If it increases at any round $t$, we increase the regularization factor $\eta$ by a fixed multiple (specifically, 2). We notice that it leads to better generalization performance over the test set. 2. The same un-regularized objective w.r.t normalized $Q^t$ is also used to perform a line search for the step size $\alpha$. For the FW update, our search space is a ball around $\frac{1}{t}$ at round $t$, while for the GA update, we search within the range $(0, 1)$.

**Training.** We use SGD with momentum $= 0.9$ for optimization. We first warm up the model with some predefined epochs of ERM (3 for COMPAS and 20 for CIFAR-10/100), followed by a maximum of $T = 5$ base models trained from the warm-up model with sample weights provided by our algorithms. Each base model is trained for 500 iterations on COMPAS and 2000 iterations on CIFAR-10/100. Each experiment is run three times with different random seeds. For evaluation, we report the averaged expected and worst-case test loss from Equation 1.

**Datasets** We conduct our experiments on three real-world datasets:

- COMPAS [Angwin et al., 2016] pertains to recidivism prediction, with the target being whether an individual will re-offend within two years. This dataset is extensively used in fairness research. We randomly sample 70% of the instances for the training data (with a fixed random seed), and the remainder is used for validation/testing.
- CIFAR-10 and CIFAR-100 are widely used image datasets. For CIFAR-10, we consider two settings: the original set and an imbalanced split [Jin et al., 2021, Qi et al., 2021]. In the imbalanced split, we make worst-case performance more challenging by randomly sampling each category at different ratios. To be precise, we sample the $ith$ class with a sampling ratio $\rho_i$ where $\rho = \{0.804, 0.543, 0.997, 0.593, 0.390, 0.285, 0.959, 0.806, 0.967, 0.660\}$. For these datasets, we use the standard training and testing splits, reserving 10% of the training samples as validation data.

**Hyperparameters** For COMPAS, we warm up for 3 epochs and then train every base classifier for 500 iterations. For CIFAR-10 and CIFAR-100, we warm up the models for 20 epochs and train base classifiers for 2000 iterations. The mini-batch size is set to 128. It should be noted that the primary aim of our experiments is not hyperparameter tuning. The experiments in this paper are designed to demonstrate use cases and compare different algorithms. Hence, while we maintain consistency of hyperparameters across all experiments, we do not extensively tune them for optimal performance.

## I.3 Interesting Observation: Boosting Robust Learners

Table 4: Mean and worst-case expected loss for RAI-FW + robust optimization algorithms.

| Algorithm | COMPAS | | CIFAR-10 (Imbalanced) | | CIFAR10 | | CIFAR100 | |
|---|---|---|---|---|---|---|---|---|
| | Average | Worst Group | Average | Worst Class | Average | Worst Class | Average | Worst Class |
| SGD ($\chi^2$) | 32.0 | 33.7 | 13.3 | 31.7 | 11.3 | 24.7 | 27.4 | 65.9 |
| RAI-FW + SGD ($\chi^2$) | 30.9 | **32.2** | 13.6 | **31.0** | 11.2 | **23.8** | 27.6 | **63.8** |
| Online GDRO | 31.7 | **32.2** | 13.1 | 26.6 | 11.2 | 21.7 | 27.3 | 57.0 |
| RAI-FW + Online GDRO | 31.6 | 33.3 | 12.9 | **24.4** | 11.4 | **19.5** | 27.8 | **51.2** |

**Boosting Robust Base Learners** We conclude our results with one interesting observation. Until now, we have been comparing our ensembles with deterministic models. As such, we acknowledge

that given the inherent differences between the two, making a fair comparison is challenging. However, we find that our setup can "boost" not only ERM but also other robust base learners i.e. if we use these robust optimization methods to find our base learners under analogous RAI constraints, we are able to further enhance the robust performance of these algorithms. The results are shown in Table 4. We hypothesize that individually robust base learners are able to help the ensemble generalize well, allowing our approach to further optimize through ensembles.

## I.4   Synthetic Datasets

In this section, we use synthetic datasets to illustrate how our RAI algorithms converge, and how different constraints on $W$ translate into performance across various *responsible* metrics. We use the following distributions to construct the datasets, and use class labels as the group labels.

- **Dataset-I:** $P(X|Y=0) = \mathcal{N}((0,0), \boldsymbol{I})$, $P(X|Y=1) = \frac{1}{3}\mathcal{N}((-3,1), \boldsymbol{I})) + \frac{1}{3}\mathcal{N}((3,0), \boldsymbol{I})) + \frac{1}{3}\mathcal{N}((0,-3), \boldsymbol{I}))$, $P(Y=0) = 0.7$, $P(Y=1) = 0.3$.
- **Dataset-II:** $P(X|Y=0) = \frac{5}{12}\mathcal{N}((-2,-2), 0.5\boldsymbol{I}) + \frac{2}{12}\mathcal{N}((-2,-2), 0.5\boldsymbol{I}) + \frac{5}{12}\mathcal{N}((2,2), 0.5\boldsymbol{I})$, $P(X|Y=1) = \frac{2}{5}\mathcal{N}((-3,0), 0.3\boldsymbol{I})) + \frac{3}{5}\mathcal{N}((3,0), 0.3\boldsymbol{I}))$, $P(Y=0) = 0.7$, $P(Y=1) = 0.3$

We sample 1000 points each for both training and testing from both distributions. Note that these datasets deliberately exhibit: 1. Class imbalance (particularly in Dataset-I) 2. Multiple minority sub-populations (within and between classes) 3. Varying noise levels in the sub-populations (predominantly in Dataset-II). Such characteristics are frequently encountered in real-world scenarios and demand responsible classifiers.

**Models**   For base learners, we use linear classifiers for Dataset-I and neural network classifiers with a single hidden layer of size 4 and ReLU activations for Dataset-II. We find that base learners can be models with varying complexity.

**Hyperparameters**   Due to the limited size of the datasets, we forgo the warm-up stage. At every round, we run 1000 iterations with a mini-batch size of 32. We run $\alpha$-LPBoost with the default value $\eta = 1$. For $\alpha$-CVaR experiments, we take $\alpha = 0.7$ across all experiments. For lower values of $\alpha$, we observe similar results and comparisons, albeit with a substantial reduction in average metrics. Consequently, we opt for conservative values to standardize average performance across all models and subsequently compare worst-case performance in responsible settings.

### I.4.1   Results and Discussion

• **Domain Oblivious (DO)**   To begin, we run ERM, AdaBoost, $\alpha$-LPBoost, and RAI games on Dataset-I. For RAI-GA and RAI-FW games, we use $\alpha$-CVaR uncertainty set as $W$. Given the class imbalance, $Y = 0$ and $Y = 1$ represent good candidates for subpopulations of interest. The results are reported in Figure 1. We immediately observe the following:

- Both proposed methods RAI-FW and RAI-GA effectively decrease the objective value and achieve lower worst-class classification loss, as compared to both ERM and AdaBoost.
- They closely follow the $\alpha-$LPBoost iterates. Intuitively, our quasi-boosting updates resemble $\alpha-$LPBoost for the CVaR objective, and that is reflected in similar objective values.

• **Domain Aware (DA)**   For this setting, we run ERM, AdaBoost, Online GDRO, RAI-GA, and RAI-FW on Dataset-II. We use Group DRO over the five gaussian groups as the uncertainty set $W$. Although Dataset-II was selected due to the presence of more pronounced subpopulation behavior, we get similar results for Dataset-I as well. The results are reported in Figure 2 and Table 5.

• **Partially Domain-Aware (PDA)**   For this setting, we run our algorithms for Dataset-II. Similar to gaussian memberships, the class labels $Y$ provide another secondary definition of implicit grouping in the dataset. We report the results in Table 5. A critical observation from the *DA* setting results is that Online GDRO, and RAI (Group) all exhibit inferior performance according to the secondary class definition i.e. although they optimize for the *known* groups (gaussian), they fail to optimize for *unknown* groups (class labels). Thus, a natural solution is to run RAI updates over the intersection

| Algorithm | Synthetic Datset-I | |
|---|---|---|
| | Average Loss | Worst Class Loss |
| ERM | 27.3 | 82.6 |
| AdaBoost | 26.5 | 27.7 |
| $\alpha$-LPBoost | 23.5 | 23.7 |
| RAI-GA | 23.9 | 24.4 |
| RAI-FW | 23.9 | 24.2 |

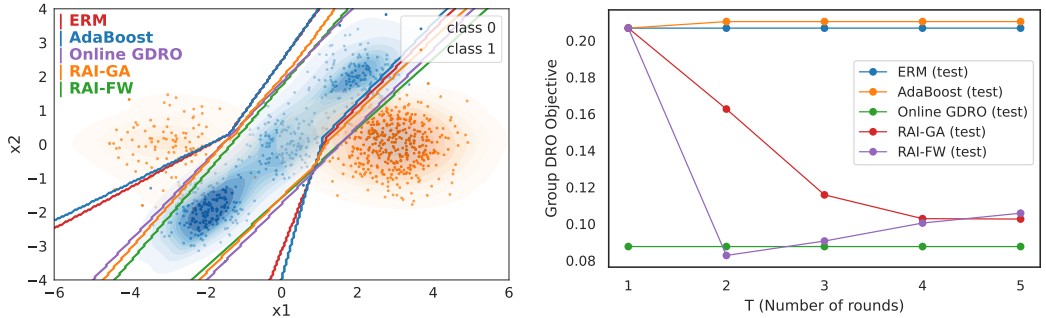

Figure 1: Results for Dataset-I. *Left:* Average and worst class losses for baselines and proposed RAI updates. *Right:* $\alpha$-CVaR objective values vs number of rounds for train and test splits

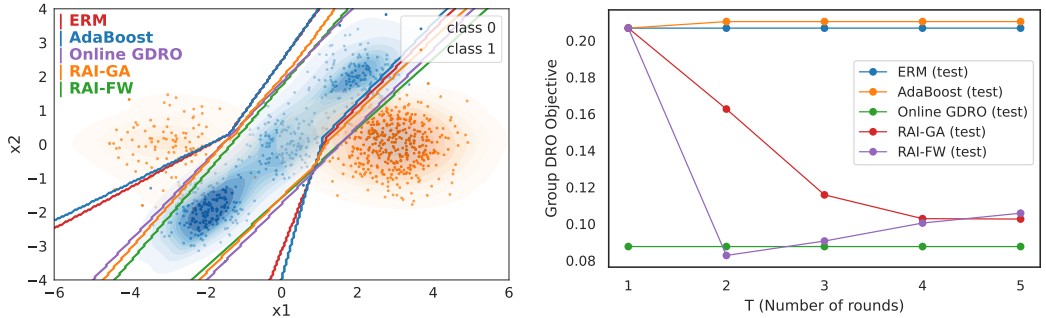

Figure 2: Results for Dataset-II. *Left*: Visualization of classifiers learned. For ensembles, we derandomized them using Definition 3. RAI methods (and Online GDRO) effectively prioritize minority and noisy instances (orange). *Right*: Testing Group DRO objective values vs the number of rounds (train values are similar and omitted to improve figure clarity). Quantitative results are reported in Table 5

of $\chi^2$ (for unknown groups) and Group (for known groups) constraints. As seen in the Table, we see that both RAI-GA and RAI-FW achieve a middle ground by significantly improving worst-case performance for both known and unknown groups.

| Algorithm | Synthetic Datset-II | | | | | | | |
|---|---|---|---|---|---|---|---|---|
| | Group 1 | Group 2 | Group 3 | Group 4 | Group 5 | Worst Group | Average | Worst Class |
| ERM | 0.0 | 3.1 | 15.2 | 22.9 | 1.6 | 22.9 | 5.5 | 5.7 |
| RAI-GA ($\chi^2$) | 3.1 | 5.6 | 10.1 | 13.3 | 2.5 | 13.3 | 5.3 | 5.9 |
| RAI-FW ($\chi^2$) | 2.1 | 4.7 | 8.9 | 13.4 | 2.9 | 13.4 | 4.9 | 5.1 |
| Online GDRO | 5.1 | 3.7 | 5.8 | 10.2 | 5.6 | 10.2 | 6.1 | 6.2 |
| RAI-GA (Group) | 5.0 | 4.2 | 7.4 | 10.3 | 3.2 | 10.3 | 5.4 | 5.7 |
| RAI-FW (Group) | 10.0 | 4.7 | 6.5 | 9.5 | 5.5 | 10.0 | 6.9 | 7.5 |
| RAI-GA ($\chi^2 \cap$ Group) | 3.4 | 4.0 | 7.3 | 10.3 | 3.4 | 10.3 | 5.1 | 5.5 |
| RAI-FW ($\chi^2 \cap$ Group) | 4.7 | 4.9 | 9.2 | 10.6 | 2.6 | 10.6 | 5.2 | 6.1 |

Table 5: Average, worst group, and worst class losses for Synthetic Dataset-II. All the algorithms listed after Online GDRO have access to group information.

