# OpenReview forum: "Responsible AI (RAI) Games and Ensembles"
_NeurIPS.cc/2023/Conference — NeurIPS 2023 poster_

### Official Review · Reviewer_KQkZ · 2023-06-24

**Soundness:** 2 fair
**Presentation:** 1 poor
**Contribution:** 2 fair
**Rating:** 5
**Confidence:** 2

**Summary:**

This paper aims to provide a general framework that is broadly applicable across responsible AI (such as ethics, fairness, train-time robustness, test-time or adversarial robustness etc). Motivated by this, the authors proposed Responsible AI (RAI) games, where a learner seeks to minimize its worst-case loss over a set of predefined distributions. This is modeled by a zero-sum game between a learner that aims to learn a responsible model, and an adversary that aims to prevent the learner from doing so. To solve this problem, the authors first go from a single predictor to a deterministic ensemble, namely a distribution over predictors that can then be de-randomized via a majority vote. Then, the authors study a relaxation of this deterministic ensemble RAI game, which they term the randomized ensemble RAI game, which can also be motivated as a linearization of the original RAI game. Finally, the authors conduct empirical analyses to demonstrate the convergence and performance of their proposed algorithms.

**Strengths:**

- The topic of responsible AI is an interesting and important area of research.
- The theoretical results are backed by proofs included in the appendix.

**Weaknesses:**

- Despite the appealing motivation of building a general framework that is broadly applicable across responsible AI, I find the proposed framework to be overly simplistic, as well as the experiments. This makes it difficult for me to connect the proposed theoretical framework with the motivation described in the introduction.
- the paper is very difficult to follow and needs to be clarified for the audiences from the general AI community.

**Questions:**

Can the authors add some more details to clarify the experiments? For example, the research questions to be tested in support of the motivation of the paper, the hypothesis, and clarify the results analysis according to these questions.

**Limitations:**

Yes the authors addressed limitation in Appendix B. I do not see any negative societal impact.

---

> ### Author Rebuttal · Authors · 2023-08-09
>
> We’re extremely grateful for the valid and insightful suggestions that the reviewer has made. Below we address some of the key concerns raised.
>
> **Scope of RAI Framework.**
>
> * The RAI framework introduced in our work aims to unify the fragmented works in the literature around RAI aspects which can be written as min-max problems. This turns out to be a large umbrella of modern problems, which includes distribution shift[1], group fairness [2], tail risk (CVaR loss)[3], etc. Our framework comes with algorithms to provably train models for these (and any combination of these) RAI aspects. As mentioned in related work, there is a large body of work that studies individual problems falling within our framework [4,5,6,7]. And we believe our framework can be very valuable in studying the optimization and generalization aspects of these problems.
> * Another important aspect of our work is that it tries to connect boosting to RAI games (to the best of our knowledge, both these lines of work have been treated separately until now), and uses the insights from boosting to design algorithms for RAI games. Please take a look at our global response where we clarify the scope of our framework and our contributions.
> * We have provided further context and summarized our contributions in the global response as well (titled *'Contributions'* and *'Importance of our developments and whether the worst-case focused developments cover all notions of responsible AI'*)
> Please let us know if any of your questions still remain unresolved. We are happy to resolve any other follow-up comments as well.
>
> [1] https://arxiv.org/abs/1911.08731
>
> [2] https://arxiv.org/abs/2011.03108
>
> [3] https://arxiv.org/abs/2110.13948
>
> [4] https://arxiv.org/abs/1610.02581
>
> [5] https://arxiv.org/abs/1810.08750
>
> [6] https://arxiv.org/abs/1806.08010
>
> [7] https://arxiv.org/abs/2106.06142
>
> **Result Section.**
>
> *  We acknowledge the discussion around the result section might not be very clear due to space constraints. Task definitions, synthetic experiments, etc. can be found in Appendix J in detail. A discussion of results can also be found in the global response (titled *'Results and Discussion'*). We will add these comments to the main paper. Here are the research questions we test and our analysis for the same.
>   * RQ1: Do Algorithms 1 and 2 converge for real-world datasets? If yes, are they able to improve worst-case performance in the considered RAI aspects from one-off deterministic baselines proposed by past literature?
>   * RQ2: Can multiple RAI aspects (e.g. in Table 1) be combined efficiently to have our model satisfy several robustness criteria at once?
>
> Here is the analysis we find:
>    * RQ1: We answer both these questions in the affirmative. RAI-FW and RAI-GA methods significantly improve the worst-case performance with only a few base learners across all datasets in all three settings, while maintaining average case performance.
>    * RQ2: We answer this question in the affirmative as well. The plug-and-play framework allows for several different $W_n$ to enhance various responsible AI qualities at once. The worst-case performance of all our baselines is a lot worse in the PDA setting than in the DA setting. This suggests that although the DA setting can optimize for the known groups, it fails to optimize for unknown groups. Thus, a natural solution is to run RAI updates over the intersection $\chi^2$ (for unknown groups) and Group DRO (for known groups) constraints. Indeed, we see that in the PDA setting, the performance lead widens, indicating that RAI is able to optimize effectively for both known and unknown subpopulations while Online GDRO suffers from some of the group information being unknown. In practice, one can construct many more novel sets $W_n$.
>
> Once again, we have also clarified some discussion around results in the main paper in our global response (titled *'Results and Discussion'*). We sincerely hope that this discussion is satisfying and addresses the reviewer’s concerns and suggestions. Please let us know if anything else needs clarification.

---

> > ### Comment · Reviewer_KQkZ · 2023-08-19
> >
> > I thank the authors for their thorough rebuttaI. After reading the response, I would like to increase my rating to 5.

---

### Official Review · Reviewer_TUyo · 2023-07-05

**Soundness:** 4 excellent
**Presentation:** 4 excellent
**Contribution:** 2 fair
**Rating:** 6
**Confidence:** 3

**Summary:**

In this paper the authors formulate a framework of responsible AI (RAI) games, which represents a variety of societal effects of AI as minimizing worst-case loss over a set of predefined distributions. The authors cover their framework and present a number of related definitions and propositions. They then present two algorithms for attempting to solve these games. They end by briefly covering results comparing their approach with a variety of baselines across a variety of common tasks/datasets.

Edit: Based on the author's rebuttal and our discussion I have increased my initial review to a weak accept.

**Strengths:**

Section 2-4 are definitely the highlights of the paper. They give a well-written argument for the utility and implications of the RAI framework. Outside of these, the clarity of the introduction and coverage of related work are also both strong.

**Weaknesses:**

There are three primary weaknesses in the current draft of the paper. First, the authors chose to use minimizing worst-case loss as a sole target for all societal effects of AI. However, this is not the only computational representation of these effects. For example, fairness is often defined in terms of symmetry across classes, which is not necessarily identical [1]. In general, this represents only one particular perspective on a subset of these societal effects. The paper would be improved with acknowledgement of this and direct discussion of what definitions are being used and why.

The second weakness is the relatively weak results. The performance of the RAI approaches appears to be very close to the baselines in all cases. This doesn't match the claim of "solving responsible AI games". The earlier text in the paper generally suggests an expectation of much stronger results. There's also an odd claim that larger models are limiting, when these are very commonly used models. Relatedly, for the third weakness, there is a lack of discussion around the results, just three bullet points and a two sentence conclusion. Reducing Sections 2-4 could allow for a better contextualization of the results to clarify if they are stronger than they appear. Otherwise, the proposed algorithms do not seem to be good fits for the RAI framework.

1. van den Brink, René. "An axiomatization of the Shapley value using a fairness property." International Journal of Game Theory 30 (2002): 309-319.

**Questions:**

1. What was the inclusion/exclusion criteria for the representations of AI societal effects not covered by the author's framework?
2. What is the reason that the performance of the RAI approaches are so similar to the baselines?

**Limitations:**

The paper does not appear to have any negative societal impacts, except perhaps overstating its contributions leading to overuse.

---

> ### Author Rebuttal · Authors · 2023-08-09
>
> We thank the reviewer for the very thought-provoking review and appreciate the time the reviewer has taken to thoroughly review our work!
>
> *  **Minimizing worst-case loss (and Q1):** There has been a surge of RAI-related topics in recent years, and it is impossible for a single framework to cover all RAI problems. However, the framework we formulate in this work can cover most of them, at the very least including all of those listed in Table 1 and their combinations. Broadly speaking, it covers RAI aspects that can be represented in a minimax fashion (for e.g. [1]). Please refer to our global response (titled *'On whether the worst-case focused developments cover all notions of responsible AI'* and *'Contributions'*) for further discussion and our full response on this aspect.
>  * **Weakness of results (and Q2):** There might be some misunderstanding regarding the claim 'weakness of results'. We believe the improvements are significant and are of similar order as related works [2,3,4]. We have now uploaded updated result tables 1 and 5  including error bars in our global response. To justify our point, we note:
>     1) from Table 1, observe that the tasks get harder from left to right i.e. the gap between average and worst-case baseline performance increases. The magnitude of improvements also increases significantly as we move to the right. For instance, RAI algorithms are able to achieve a \~5\% improvement in worst-case performance over the baselines for CIFAR-100.
>     2) the improvements in datasets like COMPAS may seem small but that is primarily because the average and worst-case performance are similar i.e. the imbalance in the dataset is intrinsically small.
>     3) Finally, we would like to highlight that good empirical results are not the only contribution of our work. One of the key goals is to initiate a principled RAI framework that encompasses a broad range of tasks that have been mostly dealt with in isolation in previous work. It also comes with algorithms and theoretical guarantees for the population risk. To further the effort of clarifying our contribution and provide a discussion on results, please refer to our global response as well (titled *'Contributions'* and *'Results and Discussion'*).
>     4) We apologize, but the comment about large models might have been misunderstood. We just underscore the fact that our RAI ensembles of smaller models are able to match up to the robust performance of the big models, which are then further outperformed by robust ensembles of those large models.
>  *  **Discussion around results** We acknowledge that there might have been a lack of clarity around some discussions around the results. A discussion of results can be found in Appendix J and in the global response (titled 'Results and Discussion'). We will add these comments to the main paper.
>  * **Algorithms** Can you kindly clarify what you mean when you say, 'the proposed algorithms do not seem to be good fits for the RAI framework'? We will be happy to resolve this question once get some more context on this.
>
> Please let us know if there is any residual feedback and we will be delighted to resolve it with further discussion.
>
> [1] https://arxiv.org/abs/2011.03108
>
> [2] https://arxiv.org/abs/2110.12459
>
> [3] https://arxiv.org/abs/1911.08731
>
> [4] https://arxiv.org/abs/2106.06142

---

> > ### Comment · Reviewer_TUyo · 2023-08-11
> > **Re: Rebuttal by Authors**
> >
> > I thank the authors for their rebuttal.
> >
> > * I understand that this formulation of min-max problems fits many cases, my request was to identify the bounds of the framework. So rather than indicating that there are many cases where it fits, I hoped the authors could indicate the cases where it does not.
> > * Apologies if I'm misunderstanding, but the results from the authors' paper seem to be much worse in terms of worst case accuracy than the results from the referenced papers.
> > * I appreciate the commitment to further discussing results
> > * Apologies, my concern here was that the proposed algorithm seemed likely to get stuck in a local maxima/minima due to the greedy nature.

---

> > > ### Author Response · Authors · 2023-08-11
> > > **Response to Reviewer TUyo**
> > >
> > > We sincerely thank the reviewer for the prompt and insightful response. Please let us know if the below discussion does not address any of the reservations held by the reviewer.
> > >
> > > **Examples not covered by the framework**
> > > 1. **Fairness:** Various notions of fairness have been studied by the ML community. While our framework captures minimax group fairness, it doesn't capture other notions of group fairness such as  Demographic Parity (https://arxiv.org/abs/1511.00830),  Equality of Odds, Equality of Opportunity (https://proceedings.neurips.cc/paper_files/paper/2016/file/9d2682367c3935defcb1f9e247a97c0d-Paper.pdf). Our framework doesn't capture individual fairness notions.
> > > 2. **Robustness** While our framework covers group robustness and certain forms of distributional robustness, it doesn't cover robustness to Wasserstein perturbations and other (un)structured distribution shifts often studied in the domain generalization community (https://arxiv.org/pdf/2203.10789.pdf, https://arxiv.org/pdf/2210.01360.pdf, https://arxiv.org/pdf/1903.06256.pdf).
> > >
> > > We will make sure to add these instances to our final revision as well.
> > >
> > > **Results**
> > >     We clarify the concern about the results here. Please note that we report classification error and not the classification accuracy.  We now discuss the baselines we compare against and the improvements we get over each of the baselines.
> > >
> > > 1. A couple of the widely cited related works actually serve as our baselines. For example, [3] is a seminal work in the DRO community. It is actually one of the baselines of our work (called Online GDRO). We outperform this baseline for both CIFAR-10 and CIFAR-100. Moreover, this can only optimize for known subpopulations within the dataset. Our framework can handle both at the same time. In the PDA setting, RAI can enhance performance in all cases. Another example, [4] considers ERM and vanilla DRO algorithms as their baselines. If we compare with ERM, our improvements are in the range 5-8\% and if we compare with $\chi^2$ DRO, we are in the similar range of 1-3\%, a similar range as the referenced work ($\chi^2$ as baseline).
> > > 2. Rather interestingly, we find that if we use these robust optimization methods to find our base learners under analogous RAI constraints (rather than ERM), we are able to further enhance the robust performance of these algorithms (across all datasets). Please refer to Table 4 and Section J.3 (Appendix) for relevant discussion.
> > > 3. Finally, these works are quite distinct in their methodology and cannot meaningfully compare against each other. Our framework is able to compete with many of them in a unified manner.
> > >
> > > **Local Optima**
> > > 1. The proposed algorithms indeed converge to a Nash equilibrium (NE) of the min-max problem. While we provide concrete theoretical guarantees for this claim in the paper, we provide more justification here. Algorithm 1 uses Follow the Regularized Leader (FTRL) and Best Response (BR) algorithms both are which are no-regret algorithms. It is well known that such no-regret algorithms when used for solving two-player zero-sum games converge to NE of the game (e.g., see Theorem 9 of http://proceedings.mlr.press/v75/abernethy18a/abernethy18a.pdf). So, as long as we can implement steps 2 and 3 of Algorithm 1, we are guaranteed to converge to a NE. Algorithm 2 is minimizing the convex objective in Equation 3. Consequently, FW and coordinate-descent style algorithms are guaranteed to converge to a global optimum of the objective.
> > > 2. Another point we reiterate is that in our bilinear games (as in RAI), there exist no local minima/maxima.

---

> > > > ### Comment · Reviewer_TUyo · 2023-08-16
> > > > **Re: Response to Reviewer TUyo**
> > > >
> > > > I apologize, I thought I'd already responded to this comment. I thank the authors for their in-depth response and reasoning! This would significantly improve the paper if these points were incorporated into it, so I'll raise my score to a weak accept.

---

### Official Review · Reviewer_yXJG · 2023-07-07

**Soundness:** 3 good
**Presentation:** 2 fair
**Contribution:** 2 fair
**Rating:** 5
**Confidence:** 3

**Summary:**

The paper demonstrates originality in its approach to proposing a general framework for Responsible AI (RAI) games. The authors introduce the concept of RAI risks and present algorithms based on game-play and greedy strategies to solve the RAI games. The connection between deterministic and randomized ensembles is also explored. Overall, the paper offers novel perspectives on addressing the challenges of responsible AI.

**Strengths:**

The paper's contribution mainly lies in providing a comprehensive framework for studying responsible AI problems and proposing algorithms for solving RAI games. By addressing various aspects such as fairness, robustness, and safety within a unified framework, the paper offers practical solutions for building responsible AI systems.

**Weaknesses:**

In general, the paper is well-structured but the overall presentation quality requires improvements, and there are instances where the language and explanations could be further improved for better clarity and comprehension.

**Questions:**

Abstract
1. “In these problems, a learner seeks to minimize its worst-case loss over a set of predefined distributions.” This sentence could be rephrased for better clarity. “In these problems” reads a bit odd, the meaning of “predefined distributions” is not very clear.
2. “In this work, we provide a general framework for studying these problems…” It is unclear what the authors mean by “these problems”. Are they referring to the aforementioned societal issues of AI such fairness, robustness, and safety? Are all RAI issues studied in this paper, or just the mentioned ones (fairness, robustness, and safety)?
3. The abstract introduces the Responsible AI (RAI) games framework but does not provide a clear definition or description of what RAI games entail. It would be helpful to briefly outline the key components and principles of RAI games to give readers a better understanding of the framework being proposed.
4. The phrase "Empirically we demonstrate the generality and superiority of our techniques for solving several RAI problems around subpopulation shift" could be rephrased to sound less self-promotional and more objective. Additionally, "several RAI problems" is too vague as RAI encompasses a broad range of issues. Furthermore, "subpopulation shift" is not mentioned or explained.

Section I: Introduction
5. The mention of fragmented work and optimization of different aspects is valuable, but it would be beneficial to provide references or briefly summarize the existing literature to give readers an understanding of the current state of research. This will help establish the context and importance of the proposed unified framework for responsible AI.
6. The authors introduce RAI games as a general framework but does not clearly define what RAI games entail. It is essential to provide a concise definition or description of RAI games to ensure readers have a clear understanding of the framework being proposed from the outset.
7. Please consider rephrasing “this is a computationally difficult game that need not even have a Nash equilibrium” for better clarity.
8. The term "population risks" is introduced without a clear context or explanation.

Section II: Problem Setup and Background
9. Appendix C is mentioned as a location for related work. But it would be beneficial to include at least a brief summary or mention of some of the related works in this section itself. This will allow readers to gain insights into the existing literature without having to refer to the appendix separately.


**Limitations:**

The authors identify the broader impact and limitations of this work in the Appendix. The limitations include inadequate evaluation, limited generalization, and the need for handling corruptions in training data.

---

> ### Author Rebuttal · Authors · 2023-08-09
>
> Thank you for the detailed review of the paper content. We acknowledge that some parts of the paper as written may not be very clear. We have provided additional clarifications in our global response. We will also edit as per all your comments in the final version.
> 1. We will rephrase the sentence for clarity in the final version of the paper. By ``predefined distributions'' we refer to the uncertainty set in RAI games.
> 2. We acknowledge that our RAI framework might not encompass all modern issues. However, it unifies a lot of subproblems that have been historically studied in isolation (Table 1) and comes with theoretical guarantees. To this end, we have clarified the scope of our contributions in our global response (titled 'Revised Contributions' and 'On whether the worst-case focused developments cover all notions of responsible AI').
> 3. By RAI games, we mean RAI problems that can be written as min-max problems. The key component in an RAI game is the domain of the inner maximization problem which captures the set of sub-populations over which we would like to have low risk. We would like to note that the domain of the inner max is predetermined and doesn't change during the course of the learning.
> 4. We apologize if the phrase sounded too self-promotional. We will edit it to be more objective in the final version of the paper. It was intended to underscore the fact that the RAI framework is not just theoretically sound, but is able to train robust models as per RAI aspects in Table 1 as well as their combinations. With only a few learners (3-5), we are able to show enhanced performance on a widely studied task over some seminal baselines, which usually have been developed in isolation and do not necessarily consider multiple aspects at once. We acknowledge that the phrase might be vague. To this end, we have extensively clarified the exact scope (and limitations) of our framework in our global response (titled 'Revised Contributions' and 'On whether the worst-case focused developments cover all notions of responsible AI'). As the task is quite well known, we had explained the task of 'subpopulation shift' in detail in Appendix J (J.1 to be specific). We acknowledge we could have placed the section better and will update the final version of our paper correspondingly. Please let us know if any of the explanation still seems unclear.
> 5. Sure, we will modify this line to: But how do we do so when the majority of recent work around these problems is fragmented (DRO [1,2]), GDRO [3], CVaR [4], Distribution Shift [5,6] and usually focuses on optimizing one of these aspects at a time?
> 6. Kindly refer to our response in point 3 above. We also have addressed this point in the global response.
> 7. Sure, we will clarify this line in our final revision. The RAI game considers fairness objectives that take a minimax representation (for e.g. [7]). In general, these games might not admit a Nash Equilibrium and thus might not be easily solvable by online learning algorithms considered. This forms one of the motivations to go from a single predictor to a deterministic ensemble i.e. to have convergence and very desirable theoretical guarantees. For more detail on two-player-zero-sum games, please refer to Appendix E.
> 8. We have introduced population risk in line 233. To reiterate, empirical RAI optimization focuses on all adversarial weightings over the given $n$ samples (drawn from $P_{data}$) which are close to the simplex. In the limit, the population counterpart is then defined over all distributions in a certain ball (defined by the appropriate distance metric) around the true data generating distribution $P_{data}$.
> 9. We will include a brief version of related work (with the most important prior work) to the main paper in the final version.
>
> We hope we have clarified all your reservations about the unclear language of the paper. We will make additional macro improvements regarding clarity in the final version. Moreover, we have thoroughly clarified our contributions, and motivation in our global response. We look forward to addressing any additional comments you might have.
>
> [1] https://arxiv.org/abs/1610.02581
>
> [2] https://arxiv.org/abs/1810.08750
>
> [3] https://arxiv.org/abs/1911.08731
>
> [4] https://arxiv.org/abs/2110.13948
>
> [5] https://arxiv.org/abs/1806.08010
>
> [6] https://arxiv.org/abs/2106.06142
>
> [7] https://arxiv.org/abs/2011.03108

---

> > ### Comment · Reviewer_yXJG · 2023-08-17
> >
> > I appreciate the authors’ thorough response and thoughtful rebuttal. I have no further comment at this time.

---

### Official Review · Reviewer_qrVx · 2023-07-27

**Soundness:** 3 good
**Presentation:** 3 good
**Contribution:** 3 good
**Rating:** 6
**Confidence:** 3

**Summary:**

This paper looks at different notions of Responsible AI (RAI) and formulates aspects such as fairness and tail-risk as weights on the samples.  The paper addresses the fact that work on RAI often only focuses on improving one of these aspects, which has been shown to worsen other aspects in some cases.  To simultaneously address these multiple aspects of RAI, the paper formulates the problem as a min-max problem, where the minimization is over an ensemble of models and the maximization is over different weight sets, with each weight set corresponding to a different aspect of RAI (e.g. subgroup performance).  The paper proposes two solutions to the min-max problem, one through game playing and alternating between maximizing the weights and minimizing over the models, and the other through a greedy descent algorithm.  The paper derives convergence bounds and contains empirical analyses showing that their method is superior to baseline methods such as Group DRO and AdaBoos for various settings, such as domain-oblivious, domain-aware, and partially domain aware.

**Strengths:**

The paper presents an original formulation to solve the problem of simultaneously addressing different aspects of responsible AI such as fairness and robustness.  This problem is interesting and relevant to the general field of algorithmic fairness and responsible AI.  Concrete examples such as those found in Table 1 help make the formulation easier to understand, and the paper does a good job of balancing mathematical proofs with empirical results to support the usefulness of the proposed solution.

**Weaknesses:**

In section 4, the introduction of deterministic and randomized ensembles could benefit from an example to help make the section easier to understand.  This section is pretty central to the problem formulation and so more wording could be devoted to this section.

For the experiments section, it did not seem that multiple RAI objectives were addressed, just subpopulation shift with various levels of awareness of the subpopulation.  It would be nice to see how the algorithm performs when more objectives are introduced.

The results table, Table 2, lacks confidence intervals, so it is hard to assess the significance of the improvements.

**Questions:**

Did the authors consider how the min-max formulation compares to treading the different RAI aspects as a multi-objective function in the loss?

How does the min-max approach compare to the baselines in terms of compute resources/time to solving the problem?

**Limitations:**

The authors discuss some of the theoretical limitations of relaxing the problem from a deterministic ensemble to a randomized ensemble, as well as the potential for improving the bounds on the convergence rate.

There did not seem to be any discussions on the practical limitations of implementing this approach.

---

> ### Author Rebuttal · Authors · 2023-08-09
>
> We are extremely thankful to the reviewer for their feedback and very insightful questions. We address them below.
>  * **Determinstic vs Randomized Ensembles:** We will add an example to the final revision of our paper to make Section 4 clearer. Here is a simple and concrete example of deterministic and randomized ensembles: consider the problem of binary classification with the hypothesis class $H$ = { $h_1, h_2$ } where $h_1(x) = 0$ and $h_2(x) = 1$ $\forall x \in \mathcal{X}$. Suppose $Q = (\frac{1}{3}, \frac{2}{3})$ is the distribution over $H$, then $h_{det;Q}(x) = 1$ $\forall x \in \mathcal{X}$ due to the majority vote and $h_{rand;Q}(x)$ is a random variable which is equal to $0$ with probability $1/3$ and equal to $1$ with probability $2/3$.
>
>
>     To add to the question, it is useful to think about why we need this distinction and when random ensembles are actually useful over deterministic ones. There are a couple of reasons: 1) the first is computational and pertains to the difficulty of solving the minimax objective  2) the second is more statistical and pertains to randomized ensembles being more powerful than deterministic ensembles. As mentioned in our work, one prime example is CVaR loss. For instance, [1] showed that for classification tasks under zero-one loss, randomized ensembles have better CVaR loss than deterministic ensembles.
>
>   * **Multiple RAI objectives in experiments:**
>     There might be some misunderstanding related to diversity in our experiments. While our experimental set is definitely not exhaustive, we indeed show experiments from multiple objectives mentioned in Table 1: $\alpha-$CVaR in Figure 1 (Appendix), and $\chi^2$, Group DRO as well as their combination ($\chi^2$ + Group DRO) in Tables 1 and 5. Similar to $\chi^2$, the usual evaluation in $\alpha$-CVaR optimization is done by evaluating worst-case group loss [2] and we find their individual performance is also very similar. Thus, we omit the $\alpha-$CVaR results from Table 1. Moreover, we are also able to display better performance than all our baselines in these cases. Note that while solving for multiple responsible AI desiderata at the same time, we have no existing formal baselines! We think it is conclusive evidence to show that our framework achieves better performance than baselines. Finally, we once again highlight the generality of the RAI framework and the impracticality of conducting experiments on all possible uncertainty $W$ sets. The framework and theoretical guarantees that come with it are the main highlights of our paper. The experiments are intended towards establishing convergence and superior worst-case performance in some of the most well-studied problems in literature (as above).
>  * **Confidence intervals:** All experiments in the original paper were averaged over three random seeds. We have now added confidence intervals to both Tables 1 and 5 as an attachment in our global response.
>
> A more detailed explanation of the setup and results can also be found in Appendix J and in our global response titled 'Results and Discussion'.
>
> [1] https://arxiv.org/abs/2110.13948
>
> [2] https://arxiv.org/abs/2106.06142
>
> * **Multi-Objective Loss Function:** One of the drawbacks of a multi-objective loss function is the huge overhead in choosing a model that achieves a good trade-off between various losses (and also the pareto optimal frontier). Even then, the solution is not guaranteed to be robust to any of the RAI aspects. We believe this is where the RAI framework provides a better formulation to work with. First of all, it allows us to treat a broad range of responsible AI scenarios under a common framework. For example, all the settings described in Table 1 and their combinations come under the RAI umbrella. Second, it is a much more computationally efficient alternative and also guarantees certain level of performance on each of the RAI aspects under consideration i.e. if we construct a DRO and CVaR uncertainty set and solve the resulting objective using our proposed algorithms, the solution will have certain robustness guarantees in both DRO and CVaR settings simultaneously.
>  * **Compute Resources/Time:** Training time scales linearly with the number of base learners. Inference, though, can be parallelized if need be. We usually find training on 3-5 learners is good enough on all scenarios explored in the paper.
>
> We are very thankful for the time spent by the reviewer to review our work and hope the discussion above clarifies the reservations the reviewer has. We will be delighted to provide any more clarifications if need be!

---

> > ### Comment · Reviewer_qrVx · 2023-08-22
> >
> > Thank you for the response.  I believe this answers my questions and will strengthen the paper.  I have no further questions or comments at this time.

---

### Author Rebuttal · Authors · 2023-08-09

## Global Response to All Reviewers
**Contributions.** We have rephrased our overall contributions to improve clarity and refine the scope of our work. We want to highlight the concrete contributions of this work below.
* Under the umbrella of “responsible AI”, an emerging line of work has attempted to formalize such desiderata ranging over ethics, fairness, train-time robustness, test-time or adversarial robustness, and safety, among others. Many of these settings (given in Table 1) can be written as min-max problems involving optimizing some worst-case loss under a set of predefined distributions. For all the problems that can be framed as above, we introduce and study a general framework, which we refer to as Responsible AI (RAI) games.
* We note that our framework encompasses not only these specific responsible AI settings but also the setting of classical boosting. Drawing upon the insights from boosting, we provide boosting-based algorithms for solving responsible AI games and derive their convergence guarantees.
* We study the population RAI risk and characterize it's global minima. Perhaps surprisingly, we show that classical Bayes optimal classifier is a minimizer of the population RAI risk for a wide range of uncertainity sets.  Finally, we initiate the study of generalization guarantees of RAI risk, which we believe is very important for understanding the test-time performance of ML models in the responsible AI setting.
* We also conduct empirical analyses to demonstrate the convergence and performance of our proposed algorithms. Owing to the generality of the framework, there is a very large body of RAI settings. Instead of attempting to exhaustively cover all of those, we  conduct experiments for one of the most well-studied problems of subpopulation shift. We demonstrate that these algorithms converge empirically and show superior performance over baselines for both synthetic and real-world datasets. Another crucial point is that all the baselines try to solve individual RAI subproblems in isolation. Instead, RAI can guarantee multiple responsible AI aspects under appropriate choices of uncertainty sets.

**Importance of our developments and  whether the worst-case focused developments cover all notions of responsible AI:**
We note that we draw our unified framework from the seminal work over the past decade by responsible AI researchers on devising non-expected risk objectives to ensure ML models are responsible. But as we note in the paper, these have resulted in a multitude of different objectives (even for a single responsible AI desideratum such as fairness), and also multiple different sub-communities (so that fairness and multiple disparate robustness communities are relatively fractured). Our goal is to draw a unified framework that combines many if not all such objectives within a single umbrella. Moreover, as we show in our experiments, such a unified framework might allow for interesting combinations such as requiring both robustness and fairness; which we imagine will result in new connections between the different communities.

We note that there is emerging work on relating worst-case performance to invariance [1]; in other words, we might be able to get approximate group invariance via minimizing an appropriately constructed worst-group risk and vice-versa. But overall we agree that RAI games might not fully cover all possible notions of responsible AI, and will be sure to emphasize this in this final version.

[1] https://arxiv.org/abs/1812.08233

**Overly mathematical developments**
We note that while at first blush the game-theoretic formulation might seem unnecessarily complex, there is an emerging trend in many of the different communities, such as tail risk, robustness, and sub-population fairness to name a few, of connecting the respective responsible AI desiderata to solving games as we note in the paper. We thus view our development as a natural teleological consequence of developments within the different responsible AI communities themselves.

**Results and Discussion**
we acknowledge that the discussion around results might be a little unclear. So we discuss it in more detail below:
 * We empirically study one of the most well-studied problems in RAI i.e. the case of subpopulation shift. We acknowledge that this is not a complete analysis, even with respect to the problems that live within the minimax framework. We aim to display convergence, plug-and-play generality of our framework, and superior performance over some seminal baselines of this task.
 * We note that for a single responsible AI desideratum, our developments mirror those emerging in the respective communities (e.g. worst-case sub-population performance) since our goal was to develop a unified framework. So the purpose of our experiments was to show that we can match baselines for a specific single desideratum (e.g. worst-case sub-population performance) while allowing for learning models that can solve multiple responsible AI desiderata at the same time, for which we have no existing baselines!

To this end, the results are shown in Table 1 (real) and Table 5 (synthetic).  As such, we can make the following observations:
  * RAI-FW and RAI-GA methods match or improve the worst-case performance with only a few base learners across all datasets in all three settings while maintaining average case performance.
  * For seemingly harder tasks i.e. a large gap between average and worst-case performance, the algorithms are able to improve significantly over the baselines. For example, we observe a $\sim$5\% improvement in performance in the case of CIFAR-100.
  * Moreover, our unified framework allows for interesting combinations such as requiring both known and unknown group fairness. We demonstrate this with the partial domain aware setting (PDA). We jointly optimize both objectives ($\chi^2$, Group-DRO) and are able to perform significantly better than baselines here.

---

### Decision · Program_Chairs · 2023-09-21

**Decision:**

Accept (poster)

**Comment:**

This paper puts forward a framework for Responsible AI (RAI) algorithms. The framework allows algorithm designers to address different aspects of RAI (such as robustness and fairness) in a unified way, via a zero-sum, or min-max optimization perspective. The reviewers have expressed several weaknesses of this work (besides of minor issues): the baselines approaches seems to perform quite well relatively to the new approach; some of you required the authors to clarify some parts of the paper (e.g., clarification on determinstic and randomized ensembles); worst-case perspective is quite limited.

That being said, in general, the reviews acknowledged the solid contribution of the framework and the extensive results the authors provide in this work, as well as the novelty of their formulation. Due to these, I support the acceptance of this work.